# Zebrafish as a Potential Model for Neurodegenerative Diseases: A Focus on Toxic Metals Implications

**DOI:** 10.3390/ijms24043428

**Published:** 2023-02-08

**Authors:** Emanuela Paduraru, Diana Iacob, Viorica Rarinca, Gabriel Plavan, Dorel Ureche, Roxana Jijie, Mircea Nicoara

**Affiliations:** 1Doctoral School of Geosciences, Faculty of Geography and Geology, Alexandru Ioan Cuza University of Iasi, No 20A, Carol I Avenue, 700505 Iasi, Romania; 2Department of Biology, Faculty of Biology, Alexandru Ioan Cuza University of Iasi, No 20A, Carol I Avenue, 700505 Iasi, Romania; 3Department of Biology, Ecology and Environmental Protection, Faculty of Sciences, Vasile Alecsandri University of Bacau, No 157, Marasesti Street, 600115 Bacau, Romania; 4Research Center on Advanced Materials and Technologies, Department of Exact and Natural Sciences, Institute of Interdisciplinary Research, Alexandru Ioan Cuza University of Iasi, No 11, Carol I Avenue, 700506 Iasi, Romania

**Keywords:** neurotoxicity, heavy metals, neurodegenerative diseases, Alzheimer’s disease, Parkinson’s disease, zebrafish

## Abstract

In the last century, industrial activities increased and caused multiple health problems for humans and animals. At this moment, heavy metals are considered the most harmful substances for their effects on organisms and humans. The impact of these toxic metals, which have no biological role, poses a considerable threat and is associated with several health problems. Heavy metals can interfere with metabolic processes and can sometimes act as pseudo-elements. The zebrafish is an animal model progressively used to expose the toxic effects of diverse compounds and to find treatments for different devastating diseases that human beings are currently facing. This review aims to analyse and discuss the value of zebrafish as animal models used in neurological conditions, such as Alzheimer’s disease (AD), and Parkinson’s disease (PD), particularly in terms of the benefits of animal models and the limitations that exist.

## 1. Introduction

Any disorder of the nervous system is called a neurological disorder. In the brain, nerves, or spinal cord, biochemical, structural, or electrical abnormalities can manifest in different symptoms. These symptoms include seizures, paralysis, poor coordination, confusion, muscle weakness, and pain [1,2]. Nowadays, neurological disorders are the most frequent debilitating diseases that affect the brain and spinal cord (the master regulators of the body). Worldwide, millions of cases of people suffering from neurological diseases—AD, PD, Huntington’s Disease, and multiple sclerosis—have been reported. An evolution in understanding the pathology of neurological disorders was noticed, signalling the need for efficient therapeutic approaches to these categories of diseases. As a result of limited animal models in research, only few drug therapies are accessible and can ameliorate or stop the progression of these neurological disorders [3].

Research models represent, in the simplest forms, the complex problems that exist in different fields, such as biology, physics, and chemistry. In science, the term model can also include non-human subjects or a few types of experimental procedures [4,5]. The animal models used for many years in neuroscience were mice and rats [4]. In addition to the classic animal models successfully used in behavioural neuroscience, a new model is emerging, with considerable advantages compared to the other models [6].

Zebrafish (*Danio rerio*, Hamilton 1822) are considered an alternative or complementary model in contrast to mammalian models. The zebrafish is a tropical fish of the Cyprinidae family [5], with a genetic structure similar to humans of between 70 and 84% and can be used foremost to study human diseases [3]. It is a model animal that has acquired popularity in behavioural studies for various neural drugs [7]. It presents several advantages compared to other animal models, including ease of handling due to its small size, higher fecundity rate, embryo transparency, and low overall costs [8,9].

These organisms are used in different scientific studies, such as toxicology, pharmacology, neuroscience, and genetics [7,10]. Researchers frequently depend on zebrafish to evaluate the toxicity of various compounds. The most recent procedures for toxicity assessment aim their attention at teratogenicity and mortality. These effects, however, from an ecotoxicological point of view, do not present the whole picture of the harmful effects that lead to a decrease in fitness and the ability to adapt to a new environment, thus, affecting entire populations. The evaluation and analysis of the results are relevant and cover aspects of toxicity at different levels of examination [11].

This fish immediately became a basic model for the neurobehavioral examination of vertebrates at the molecular level. Zebrafish were the basis for the diversity identification of genes that act negatively in distinctive aspects of neuronal development and function [5]. It is currently the essential model for studying neurobehavioral aspects relevant to humans [7,10]. The neurobehavioral function and life of model organisms are affected by numerous toxic substances whose impact is more accessible to evaluate in the laboratory than in their living environment. Diverse behavioural experiments (aggression, anxiety, and exploration) are carried out in the laboratory for a better understanding of the impact of contaminants because these aspects are closely related to traits that influence life course, such as sociality, defence, and reproduction (Figure 1). The effects of neurotoxicants are analysed, focusing on embryos and larvae as well as adult zebrafish, demonstrating that these behavioural domains are very susceptible, even at low concentrations of the substances. These strategies are exemplary in ecotoxicological research using zebrafish as aquatic models [11].

Zebrafish are an increasingly used model organism to study the neural basis of behaviour and disease because of their genetic similarity to humans and the ease with which their brains can be analysed [3]. Certainly, zebrafish brains have distinctive traits compared to those that exist in the mammalian brain [12], although the overall configuration of the cerebral cells and brain considerably resembles both models [3]. For example, zebrafish brains do not have a cortex or a hippocampus, instead, they own equivalent formations, which adequately function similarly to the mammalian brain, thus, allowing for comparable behaviours with mammals in an experimental setting. The absence of the substantia nigra and ventral tegmental area in the zebrafish brain does not impede the zebrafish from exhibiting behaviours typically assigned to those components and being susceptible to neuropharmacological manipulations [12].

Studies have shown that the basic organisation of the zebrafish brain, including the formation of neural circuits, is comparable to that of the human brain. In particular, zebrafish have a well-developed cerebellum, a brain region involved in motor coordination and learning, which is analogous in function and structure to the cerebellum in humans [13]. In addition, zebrafish have a variety of neurotransmitter systems, including dopamine, serotonin, and gamma-aminobutyric acid (GABA), which are also present in the human brain [14]. The habenula, which controls the delivery of neurotransmitters, such as dopamine and serotonin, has maintained its structure through evolution, therefore, producing similar data referring to behaviours in zebrafish and humans. Scientific experiments revealed similarities between rodent models of depression and stress-related behaviours in zebrafish regarding the hyper-activation of the habenula [3]. These similarities in brain structure and function make zebrafish a valuable model for studying neurodegenerative diseases, such as AD and PD, which affect the same neurotransmitter systems in humans [14]. The usefulness of the zebrafish as a model comes into question when alternative neurotransmitter complexes are involved (notably for nitric oxide and histamine) because they are not completely distinguishable in zebrafish [12].

Zebrafish also have the advantage of being transparent and having short-timed generations, making them useful for studying disease progression and screening for therapeutic potential. However, there are also significant differences between the brains of humans and zebrafish, and the complexity of the human brain means that it is challenging to reproduce all the features of human neurodegenerative disease in an aquatic animal [15]. Although using zebrafish in neurodegenerative disease studies presents many advantages, there are also some limitations and disadvantages to consider. One limitation is that zebrafish do not develop complex brain structures or behaviours that are present in mammals, such as mice and rats, which may limit their usefulness for studying certain aspects of neurodegenerative diseases [16]. On the other hand, zebrafish do not develop age-related neurodegenerative diseases, such as AD, PD, or cerebrovascular disease, which may limit their usefulness for studying the aging-related aspects of these diseases. In addition, zebrafish have a relatively simple nervous system compared to mammals, which may limit their usability for studying complex neural circuits and behaviours. Finally, some methods used in zebrafish studies may not be directly translatable to humans because there may be differences in the underlying disease mechanisms [17]. Predominantly, scientists are in consensus that zebrafish are acceptable models for experimental studies in mammalian neurobiology [12], particularly the functioning of the human brain [3].

The outline of this present review highlights the usefulness of zebrafish models for analysing the changes in behavioural patterns, biochemical responses, and histological structures of the zebrafish when exposed to heavy metals (cadmium—Cd, mercury—Hg, and lead—Pb), which enable the comprehension of neurodegenerative disease (AD and PD) and their progression mechanisms. We argue that zebrafish models of complex brain disorders are developing into a rapidly emerging critical animal model in the field of neurotoxicological research.

## 2. Methodology

The literature databases searched for information in the present manuscript until inception (November 2022) were ScienceDirect, PubMed/MEDLINE, Web of Science, Scopus, and Cochrane Database of Systematic Reviews (CDSR).

Several keywords, such as “neurotoxicity”, “cadmium”, “mercury”, “lead”, “heavy metals”, “neurodegenerative disease”, “Alzheimer’s disease”, “Parkinson’s disease”, and “zebrafish”, were used during the database search. We selected all relevant literature based on the title, abstract information, and full content.

Conference posters, letters to the editor, or computational simulations were not considered suitable. English-written articles were the focus of our analysis. Three authors (E.P.; R.J.; V.R.) independently inquired about the data, and common consent with the remaining four authors (D.I.; G.P.; D.U.; M.N.) solved any existing differences.

## 3. Results

### 3.1. Neurotoxic Effects of Heavy Metals

Since prehistoric times, people have integrated parts of various metals into different pigments of paint. Metals, such as chromium (Cr) and arsenic (As), were gradually eliminated from the production process of green and red dyes because of their toxicity. Cd, well known as a carcinogenic, is still used in manufacturing yellow, orange, and red pigments. Taking into account its wide range of uses in pipes, paint additives, ceramic glazes, and cosmetic products, and its use even as a wine sweetener, Pb is quite challenging to eliminate. The most widely known toxic metal is Hg, which doctors have used as a panacea for various ailments, starting with constipation and syphilis treatments [18]. Aluminium (Al), a less toxic metal, can be removed from the body through regular elimination activities, although other metals reach the food chain and are bioaccumulated, generating a chronic nature [19].

Raikwar et al. [20] reported on heavy metals that organise into four main categories based on their relevance to health (Figure 2). Essential metals called micronutrients—copper (Cu), zinc (Zn), cobalt (Co), Cr, manganese (Mn), and iron (Fe)—are toxic if they exceed the recommended limit. Non-essential metals include barium (Ba), lithium (Li), and zirconium (Zr). Tin (Sn) and Al are considered lesser toxic metals. In addition to being called trace elements, Cd, Hg, and Pb are highly toxic heavy metals [20].

Neurotoxicity is a well-used term to describe a multitude of adverse neurological effects. Exposure to various biological substances and physical or chemical agents can affect the functions or structure of the central nervous system (CNS) [21]. Neurotoxic agents can induce neurophysiological transformations. These changes include cognitive or motor symptoms and mental or behavioural disorders [22]. Synthetic pesticides, drugs, bacterial and animal neurotoxins, and heavy metals can be neurotoxic agents. The toxicity of heavy metals, unlike other environmental contaminants, has been known since ancient times and has been extensively analysed and investigated [21], though a few aspects that require our attention on this critical topic remain.

Distinctive public health measures have been undertaken to prevent, control, and treat heavy metal toxicity that occurs from accidents, the consequence of environmental factors, or occupational exposure [19]. Unfortunately, the population around the world remains unprotected from highly toxic heavy metal exposure. The cause of excessive exposure originates from diverse sources, such as industry, which discharges harmful heavy metal waste into the environment [23]. Exposure to metals can be long term (chronic exposure) or short term (acute exposure), and their toxicity depends on the amount absorbed and the route of exposure. It can result in numerous disorders that lead to considerable damage due to a higher incidence of oxidative stress (OS) [19]. Cd, Hg, and Pb (considered pollutants found in the soil, air, and water) are linked to adverse impacts on overall health [24,25]. They primarily affect the CNS and the peripheral nervous system (PNS), along with different organs—lungs, liver, and kidneys [26].

There is a growing association between exposure to heavy metals and neurodegeneration. It is a concern for public health because of the increasing prevalence of dementia, the negative consequences of neurodegeneration-related disabilities, and increasing environmental pollution [27]. Exposure to highly toxic metals (Cd, Hg, Pb) induces neurological deficiency in the human body, including memory loss, paralysis, mental disorder, and paraesthesia. Neurotoxicity results in neurodegenerative disease (Figure 3) [28]. Studies have consistently found no significant estimates of hazard risk for AD or PD, partly for limitations in accurately diagnosing these conditions and assessing chronic exposures [27]. Researchers believe experimental inquiries focus predominantly on the cellular and molecular mechanisms that induce neurotoxicity. Remarkably, analyses have targeted specific pathways, but limited reports spotlight the systemic effects on the brain [29]. Aside from proteomic and functional analysis, precise pathway evaluation may generate new perspectives on cellular pathways and damaged functions by different toxicants, including metals [30]. Contemporary scientists conducted experiments to investigate the connections between various pathways, including inflammation, mitochondrial dysfunction, apoptosis, autophagy, and OS, to bring a profound understanding of neuronal deteriorating processes and heavy metals’ implications [31]. However, experimental data suggest that exposure to heavy metals may increase the risk of neurodegenerative diseases by causing OS, which leads to hallmarks for both AD and PD, such as the formation of protein aggregates and neuroinflammation. Increasing the production of reactive oxygen species (ROS) and deregulating antioxidant enzymes, heavy metal exposure can lead to the formation of protein aggregates, such as tau, amyloid beta (Aβ), or alfa-synuclein (α-syn), overwhelming the body’s degradation systems. This chain activates neuroinflammation and further increases OS, creating a self-perpetuating cycle that leads to neuronal loss in specific brain regions, such as the hippocampus and cerebral cortex in AD and substantia nigra in PD. The OS caused by heavy metals also affects signalling pathways, leading to the accumulation of toxic materials in neural cells, such as damaged/aberrant proteins, Aβ in AD or α-syn in PD, oxidative byproducts, or the oxidation of deoxyribonucleic acid (DNA), which can alter genetic or epigenetic regulation [27].

Several studies support the link between Cd, Hg, and Pb exposure and neurodegenerative disorders [32,33]. Scientists observed in various experiments that in cellular and animal model systems, heavy metals, including Cd, Hg, and Pb, contribute to the pathologies of AD and PD [32]. Toxicological analysis has pointed out that post-mortem patients suffering from neurodegenerative diseases had higher concentrations of Pb and Cd in their cerebrospinal fluid [34]. Likewise, epidemiological studies have linked Pb exposure to cognitive decline, dementia, and AD [32]. Hg also plays a role in these disorders, with researchers finding elevated Hg concentrations in the locus coeruleus neurons of AD patients [34,35]. Furthermore, Pb and Hg can induce OS in brain endothelial cells, which can contribute to neurotoxicity [36]. On the other hand, Pb and Cd exposure can also affect astroglial plasmalemma glutamate transporters, leading to neurodegeneration [37]. In addition, exposure to Cd and Pb can affect the activity of several brain enzymes, which are neurotoxicity markers [36,38]. Thus, understanding the relationship between early life environmental factors and neurodegenerative diseases is becoming increasingly important [27].

#### 3.1.1. Cd-Induced Neurotoxicity

One of the heavy metals still used in several processes is Cd, which can cause concern because of its effects on people’s health through contaminated water sources and soil. The International Agency for Research on Cancer (IARC) catalogued Cd as a carcinogenic factor [39]. Cd is a risk factor for numerous conditions, such as osteoporosis, cardiovascular, cerebrovascular diseases, and nephrotoxicity. Cd exposure is a concern because of its neurotoxicity, though its mechanism of action is not fully understood [32,40].

According to Yesildag et al. [41], Cd impairs redox homeostasis through a removal process of cations occupying different antioxidant enzyme sites, binds to sulfhydryl groups of proteins, including antioxidant enzymes, and blocks them. Moreover, Cd deteriorates cellular glutathione, GSH (an essential antioxidant in the body), and, finally, leads to the accumulation of ROS and deterioration in oxidant and antioxidant balance [41]. Furthermore, Cd induces neurotoxicity through ROS activation of the mammalian target of rapamycin (mTOR) pathways. Rapamycin is a neuroprotective macrocyclic lactone that prevents neuronal cell death. Antioxidants could be administered to prevent Cd-induced neurodegenerative diseases, since redox-based signalling mechanisms coordinate the mTOR complexes. While not inhibited, Cd-induced apoptosis diminished RICTOR-silenced PC12 cells, emphasising the relevance of this regulator concerning heavy metal neurotoxicity [30].

Observations made during toxicological assays reveal an association between Cd exposure and neurodegenerative diseases through different pathogenic mechanisms, such as apoptosis, neuroinflammation, OS, modifications in blood–brain barrier (BBB) permeability, Aβ aggregation, and tau neurofibrillary tangle production [32]. Cd can pass the BBB and reach the CNS. The BBB protects against potentially toxic substances that may accumulate in the brain [42]. Therefore, Cd-induced neurotoxicity initiates pathogenic processes that conclude with a deterioration at the cognitive level and AD neuropathology. Using the olfactory system to avoid the BBB, Cd instantly reaches the CNS, thus, inducing a continuous and permanent impairment in adult neurogenesis in the hippocampus and olfactory bulb [32]. Likewise, Cd exposure is partly responsible for PD and other afflictions, such as learning difficulties, olfactory dysfunctions, peripheral neuropathy, and memory deficit [42]. Zebrafish’s exposure to high levels of Cd throughout initial embryonic growth provoked abnormalities in apoptosis and neural tissues [43]. Following in vitro treatment with a 4 µM Cd concentration, Deng et al. [42] discovered a severe apoptosis cellular process in mice embryonic neuronal cells by evaluating the status of activated Caspase-3 (a known indicator for cell death) [42]. Current investigations have revealed that Cd can cross the placenta during neonatal development when the CNS is notably susceptible to pollutants. Moreover, Cd has been identified in lactation milk, thus, propagating the negative effect on a newborn’s neurodevelopment [42,44].

#### 3.1.2. Hg-Induced Neurotoxicity

Hg is considered a highly toxic non-essential element whose presence in biological systems, even in low concentrations, is harmful. Chronic and acute exposure have similar adverse effects, characterised by renal, immune, and cardiovascular system diseases. Thus, a direct link between the detrimental effects of Hg and infants’ neurological development has been established [22]. Distinct examinations recorded significant associations between methylmercury (MeHg) exposure and neurobehavioral impairment in young children. Novel research reveals evidence of Hg effects focusing on prenatal exposure to MeHg. Populations with a higher consumption rate of freshwater fish and seafood have been examined for different degrees of MeHg contamination. Results obtained by a group of scientists in a New Zealand study display a reduction in three points in children’s intelligence quotient (IQ) because of their mothers’ Hg concentration in hair being higher than 6 µg·g^−1^. A subsequent investigation in the Faroe Islands connects impairment in children’s attention, memory, language, and visuospatial perception with MeHg exposure. On the other hand, another ambitious study in the Seychelles offered no support for Hg prenatal neurotoxicity, subsequently rectifying postnatal exposures in the fish-eating community [45]. In addition, people vastly dependent on products based on fish and seafood face a greater risk of Hg poisoning, unlike people who rarely consume fish [46].

A neurological illness among people from Minamata City, “Minamata disease”, was induced by the consumption of fish and shellfish polluted by anthropogenic emissions and the release of MeHg in the Minamata Gulf. Among the symptoms of this disorder are defective motor functions, the uncontrollable motion of arms and legs, flawed verbalisation, and impaired hearing and vision. This ecological disaster prompted governments from 128 countries to sign the Minamata Convention in 2013, to create a legislative precedent in safeguarding the environment and human health against irresponsible industrial chemical disposal of Hg and Hg compounds [47].

It is well known that neuronal MeHg exposure generates adverse effects in vivo and in vitro, resulting in ROS overproductions, leading to protein injuries, nucleic and lipidic acid, and, eventually, cell apoptosis [48,49,50]. Furthermore, MeHg, in addition to boosting ROS overproduction, diminishes antioxidant species and induces mitochondrial dysfunction. Various effects can be associated with MeHg, such as apoptotic death, perceptible cell biochemical and morphological changes, alterations in mitochondrial permeability, chromatin condensation, and cytochrome release [30,31]; one of the most noticeable apoptotic biomarkers that have a fundamental role in the initiation and execution of cell death is caspase cascade activation [51]. Thus, 2 h exposure to 3 μM MeHg of Neuro-2a cells increased caspase-3 activity significantly. Both in vivo and in vitro, MeHg induces gene expression in neurological adverse reactions [51].

The quantitative proteomics analysis highlights the mechanisms of MeHg-induced neurotoxicity in Atlantic cod (*Gadus morhua*, Linnaeus 1758), such as OS, mitochondrial dysfunction, and changes in calcium (Ca) homeostasis [52]. Likewise, scientists discovered similar pathways in the toxicological research of mammals exposed to MeHg, which had impaired biological functions [30]. Being a widely known environmental toxicant, Hg can provoke alteration in the zebrafish’s cell physiological mechanisms. Exposure to a specific Hg compound (MeHg) causes an imbalance of proteins, Ca homeostasis, and enzyme malfunction by interaction with sulfhydryl groups [30,31].

Dr Pigatto and his research team [53] commend Dr Chakraborty for his Commentary regarding the revelation of a potential link between a low dose for a long duration of Hg exposure and AD prevalence in India’s human population. Non-essential metals operate through molecular ionic mimicry, acting as substitutes for essential metals in the rivalry towards metabolism in biological systems [53]. The latest scientific literature establishes a connection between Hg deposits in the amygdala [54] and hippocampus [55,56] (cerebral areas associated with memory and additional cognitive functions) and the certainty that Hg has a potential role in the pathophysiology of AD [26,53,57,58].

#### 3.1.3. Pb-Induced Neurotoxicity

Pb exposure can result from anthropogenic activities, such as smelting, mining, the leisure industry, cement and paint use, and manufacturing batteries. In animal exposure, Pb can cause various toxic effects on behavioural, physiological, and biochemical functions. It can damage the CNS, PNS, cardiovascular system, haematopoietic system, and other organs, such as kidneys and liver [59]. The complex mechanism of Pb-induced neurotoxicity is not yet fully understood. Neurotransmission decline, OS, disruption in cellular signalling, and changes in membrane biophysics are the main hallmarks of Pb-induced neurotoxicity. Furthermore, it can cause OS straightaway or through lipid peroxidation, favourable to the generation of ROS, including hydrogen peroxidation, singlet oxygen, and reduction in antioxidant resources [45].

Prevalent articles report the fact that metals may produce toxic effects targeting mitochondria. Mitochondria are considered the primary source of cellular energy metabolism and were intensely investigated in connection with xenobiotic-induced neurotoxicity. Pb, a non-redox-active metal, determines an imbalance in mitochondrial homeostasis, mostly through GSH diminishment, because of thiol groups’ suppression, reduces antioxidant enzyme activity, and changes the integrity, functioning, and permeability of membranes, allowing lipid peroxidation to occur. Thus, cell death can occur when mitochondrial components are directly damaged and may disrupt homeostatic mitochondrial functions [60]. Pb inhibits energy metabolism by stimulating protein kinase C in capillary cells and restricting sodium–potassium pump activity. At a cellular level, Pb prevents mitochondria from discharging Ca, forcing the development of permeability transition pores and programmed cell death mechanisms, leading to mitochondrial self-elimination processes [61].

In the human cell model, Pb generates a decline in neuronal differentiation [62]. Zhou et al. [63] experimented with rats regarding the effects of Pb exposure. The outcome of these laboratory tests reveals that Pb exposure in young rats increases the accumulation of Aβ and deposition of amyloid plaques, increases amyloid precursor protein (APP) expression, and disrupts brain cholesterol metabolism [63]. Likewise, Pb exposure induced tau hyperphosphorylation and neuroinflammation in rodents and Aβ deposition in non-human primate models’ brains. In zebrafish, Pb alters gene expression and may be associated with an expanded risk for AD [64,65,66]. Pb exposure in adult rodents affects axonal myelination, reduces the size of prefrontal cortex grey matter, and promotes pathological early-life apoptotic neurodegeneration [62].

Long-term Pb exposure interferes with Ca signalling cascade pathways and calmodulin kinase regulators, which take part in memory consolidation and synaptic plasticity [64,67]. Particularly during the neonatal period, Pb generates processes that rely on calmodulin, usually activated by Ca and Zn [61]. Frequently, Pb-induced toxicity outcomes in the early stages of zebrafish life include loss of axonal density and obstruction of cerebral transcriptomic pathways’ growth, neurogenesis, axonogenesis, and CNS progress [64,67]. Transgenerational changes in brain transcriptome may occur in zebrafish exposure to Pb^2+^ during developmental stages. CNS morphology can suffer developmental alteration through different mechanisms, such as BBB disruption, alterations in synaptic proliferation, pruning interference with thyroid hormone transport, and shifts in neurotransmitter release [62]. Therefore, the dopaminergic (attention and functioning), glutamatergic (learning and memory) [62], and GABAergic systems’ operations are affected, and N-methyl-D-aspartate-ion channel obstruction occurs [61]. After conveniently crossing the BBB because of its similarity to Ca ions, Pb causes damage to the prefrontal cerebral cortex, hippocampus, and cerebellum. These effects consequently lead to wide-ranging neurological diseases, including nerve damage, brain deterioration, mental delay, behavioural issues, and probably even schizophrenia, PD, and AD [61].

### 3.2. Zebrafish as a Research Model for Heavy-Metal-Induced Neurological Disorders

Contemporary studies indicate that zebrafish can be an essential model because of their high physiological homology with humans, susceptibility to genetic manipulation, and pharmacological sensitivity [68]. According to Green and Planchart [69], zebrafish is an important model for understanding neurodegenerative diseases, contributing to the progress of new transgenic lines that would heighten interest in researching these disorders [69]. The zebrafish is a significant model organism for studying the therapies and pathology of neurodevelopmental disorders (AD and PD) [68]. Symptoms in zebrafish are similar to those in humans [70].

Zebrafish experiments can prompt the extrapolation of the neurotoxic effects of chemicals on humans by using zebrafish as a model system for toxicological estimates that anticipate chemical hazards [71]. Scientific experts believe environmental exposure to heavy metals is linked to negative issues related to cognitive impairments and neuropathological damage [53] (Figure 4).

#### 3.2.1. Zebrafish as a Model for Heavy-Metal-Induced AD

Worldwide, the predominance of neurodegenerative diseases is constantly increasing [32,73]. AD, which was discovered for the first time in 1907 [16], is the most common form of dementia in the elderly [47,55,74]. Changes in AD patient pathology are expressed by massive neuronal loss and impaired synaptic processes in the cerebral cortex, especially in the temporal and frontal lobes and hippocampus [75,76]. Genetic mutations cause, according to [16], less than 1% of AD cases, while the rest are associated with genetic susceptibility, environmental factors, and other factors. The primary symptoms of AD include cognitive dysfunction, memory loss, and accompanying conditions, such as anxiety or depression. The pathological changes in AD include neuronal cell death in brain regions that control cognitive functions, extracellular Aβ plaques, and intracellular neurofibrillary tangles. Researchers have proposed several potential causes for AD, including the amyloid cascade hypothesis, tau hyper-phosphorylation, neuroinflammation, cholinergic hypothesis, and metal ion hypothesis [16].

Elevated values of the Aβ peptide in cerebrospinal fluid are among the first indicators of early- and late-onset AD [10]. Therefore, scientists focused on elaborating the Aβ hypothesis based on the Aβ plaque presence in the brain. They suggested Aβ, including OS and inflammation, plays a crucial part in activating the disease’s neuropathology. Aggregation of Aβ and deposition for a period were considered the primary mechanisms of the disease pathology. Nevertheless, the Aβ hypothesis is the key to the disease and has to assert other distinct interpretations [77]. When the disease is discovered because of the symptomatology, it is already too advanced to present an effective response to therapeutic interventions represented by drugs that inhibit Aβ plaques and stop cognitive degradation. Therefore, it is necessary to identify and extend new pharmaceutical strategies for this disease [10]. The development and progression of AD is a multifactorial process that involves different molecular pathways. Inflammation and oxidative damage are critical features of brain atrophy in neurodegenerative diseases, particularly in AD. Metal toxins, such as heavy metals, are neurotoxic and can cause a cascade of neurotoxic effects, including an increase in OS, inflammation, and Aβ aggregation [55].

AD has two main pathological features in the brain: increased quantities of extracellular Aβ protein plaques [46] and intracellular aggregates of tau hyperphosphorylation [54]. Neurofibrillary tangles are an essential component of these hyper-phosphorylated tau proteins and are assumed to be one of the AD hallmarks. Neurofibrillary tangle bundles, found in the hippocampus and neocortex, are composed of an abnormal structure of tau protein that disrupts the microtubules, which provide support and nutrients to neurons, thus, leading to neuron degeneration [46]. Previous research has shown that neurofibrillary tangle deposition is actively linked with cognitive decline in AD compared with abnormal Aβ deposition because neurofibrillary tangles are directly involved in the induction of neurodegenerative processes [72]. Studies have implied that exposure to heavy metals, such as Hg, Pb, and Cd, can increase the phosphorylation of tau and the formation of neurofibrillary tangles, further contributing to AD pathology [54].

Thawkar and Kaur [72] mentioned that apoptosis, synaptic loss, microglial activation, and memory deficits are among the processes generated through Aβ42 aggregation in adult zebrafish neurons [72].

Experimental research guided by Lee and Freeman [64] aimed to investigate the impact of Pb exposure (10 mg·L^−1^ Pb) during embryonic stages on gene expression in the brains of adult zebrafish, with a target on the differences between males and females. Additionally, the study found that female zebrafish had changes in the expression of several genes (amyloid precursor protein—*APP*, apolipoprotein E—*APOE*, and sortlin-related receptor precursor—*SORL1*) related to AD. Furthermore, *APP* produces a protein Aβ that can accumulate in the brain and contribute to AD. *APOE* is a gene linked to an increased risk for AD, and *SORL1* is a recently identified genetic risk factor for the disease because it is assigned to diminish Aβ production. Likewise, *APOE* is also involved in the transport of cholesterol in the brain, which is essential for cell membrane formation. The expression of an enzyme, which contributes to the processing of APP and the production of Aβ, was significantly altered in female zebrafish. Overall, the study supports the idea that developmental exposure to Pb at low concentrations can have long-term effects on the nervous system [64].

The tangles hypothesis is a prevalent theory about the progression of AD. There is a strong correlation between the presence of tangles and the status of the disease. However, the amyloid hypothesis is essential, as mutations in tau do not lead to plaque deposition [78]. In AD, the tau protein, widely expressed in neurons, is involved in the formation of microtubules [72]. Mutations in microtubule-associated protein tau (MAPT) may encode the tau protein associated with frontotemporal dementia [69] and alter its ability to interact and bind with microtubules, leading to a tauopathy process [72]. An increase in neurofibrillary tangle formation and a decrease in tau binding to microtubules can occur when an ordinarily human tau mutation, A152T tau (comprising a single G > A nucleotide change), manifests [69]. The inactivation of sirtuin 1 (SIRT1), an enzyme, leads to the post-translational alteration in tau, specifically acetylation of tau, which acts as a precursor for tau phosphorylation and tauopathy. Hyperphosphorylated tau protein dissociation from the microtubule results in microtubule destabilization. This process includes hyperphosphorylation and aggregation of MAPT, which, in abnormal quantities, leads to the toxic formation of neurofibrillary tangles in the brain [72]. The formation of extracellular plaques and intracellular neurofibrillary tangles are the neuropathological hallmarks of AD [57]. Studies on neurons have shown that among Cd, Pb, and Hg, only Hg at low concentrations induces the hallmark of abnormal tubulin aggregation seen in AD [46]. Zebrafish induced with the A152T-tau mutation present proteasomal deficiencies and neurodegeneration. Pharmacologically, autophagy upregulation could partially improve the expression of these effects [69].

A hypothesis regarding AD prevalence pointed out that the deterioration in the synthesis of the neurotransmitter acetylcholine (ACh) contributes to the manifestation of AD [16]. In AD patients, a cholinergic transmission decrease is responsible for abnormalities in the functional and cognitive domains. Eventually, novel research demonstrated that cholinergic dysfunction causes cognitive impairment only in an indirect manner [78]. The cholinergic system, which uses ACh as a neurotransmitter, plays a role in cognitive processes by initiating certain receptors’ responses. Acetylcholinesterase (AChE) and butyrylcholinesterase (BuChE) enzymes are responsible for maintaining ACh levels [79]. These enzymes, which degrade ACh and are implicated in the control of the cholinergic system, contribute to AD progression [16].

Studies have confirmed the existence of choline acetyltransferase (ChAT) and AChE in the zebrafish brain [72]. Furthermore, histochemical [72] analysis of zebrafish brain tissue established the presence of AChE [72,79]. This enzyme is a biomarker for heavy metal accumulations in zebrafish and plays a role in neurodegenerative disorders, such as AD. Although zebrafish possess only the AChE gene, which is accountable for the entire ACh degradation process, it still is a suitable model for studying the effects of heavy metal toxicity on the brain. Hg chloride (HgCl_2_) has been shown to reduce antioxidant competence and inhibit AChE activity, suggesting that oxidative damage in the zebrafish brain may be involved in the neurotoxicity promoted by heavy metals. Alterations in neurotransmission systems can explain some neurotoxic characteristics of heavy metals, and identifying biological alterations related to the cholinergic systems during metal exposure may provide important insights into the neurochemical and molecular targets involved in neurotoxicity. Overall, heavy metals target different levels in the body, including enzymes involved in the control of cholinergic transmission and antioxidant competence, which are a small portion of the wide range of actions promoted by these pollutants [79].

A group of scientists led by de Lima [80] examined the impact of Cu, Pb, Fe, and Ca on the activities of AChE and carboxyl esterase (CbE) in zebrafish. In vitro results illustrated that AChE was significantly inhibited by high concentrations (10 and 20 mmol·L^−1^) of Cu, Fe, Pb, and Cd, while CbE was inhibited only at 20 mmol·L^−1^. In vivo findings revealed that Pb and Cd caused AChE inhibition at high concentrations, Fe had no effect, and Cu increased AChE activity at a 0.06 mg·L^−1^ concentration. These results highlight the significance of metals as cholinergic inhibitors in the zebrafish nervous system [80].

Exposure to Pb chloride (PbCl_2_) impacted serotonin and AChE expression in zebrafish. Low brain serotonin levels correlated with the reduced locomotor activity seen in PbCl_2_-exposed fish since serotonin regulates movement in model animals. ACh plays a crucial role in memory and nerve muscle function. AChE, the enzyme collapsing ACh in the brain, is usually boosted by contaminants, such as heavy metals, pesticides, and insecticides. In this study, scientists found that 30 days of low-concentration PbCl_2_ exposure reduced AChE activity. However, increased AChE activity has also been reported in fish after heavy metal exposure [81].

Another study reveals that Cd exposure boosts cell death in cholinergic neurons, leading to alterations in AChE and the degeneration of basal forebrain cholinergic neurons [82]. Memory deficits seen in AD are associated with the loss of cholinergic neurotransmission because of the degeneration of cholinergic neurons in the basal forebrain [32].

The GABAergic system is widely present throughout the brain and plays a significant role in balancing excitatory and inhibitory signals. The equilibrium between these signals is crucial for the simultaneity of various CNS functions [83]. GABA is an inhibitory neurotransmitter that regulates neural functions by modulating the activity of postsynaptic cells. GABA neurotransmission deficiencies play a vital role in CNS disorders [84]. Preclinical and clinical studies have established that dysfunction of the GABAergic system leads to an imbalance in excitatory and inhibitory signals, which is a potential marker for the early stages of AD [83].

Studies have documented the existence of GABAergic neuron receptors (in the telencephalon, hypothalamus, tectum striatum, and olfactory bulb) and genes similar to human glutamic acid decarboxylase in zebrafish and observed their early-stage expression. This model has been extensively validated for research on neuronal pathways analogous to the human brain, such as the cholinergic, glutamatergic, and GABA pathways, which play a role in the manifested behaviour of the zebrafish. The administration of heavy metals can change neurotransmission pathways by targeting various molecular mechanisms [85].

Wirbisky et al. [86] exposed zebrafish embryos to 10, 50, or 100 ppb Pb to analyse the effects of Pb on the GABAergic system during development. Morphological alterations, tissue uptake, neurotransmitter levels, and gene expression were analysed. Analysis revealed that alterations in gene expression throughout the GABA levels and GABAergic system were doses and developmental-time-point specific. Results prove that exposure to low concentrations of Pb affects the developing CNS, with a focus on the GABAergic system [86].

In an experiment guided by Cambier et al. [87], fish brains accumulated a significant amount of Hg (30.2 ± 4.2 lg·g^−1^ dry weight) after 25 days of exposure to 3 nmol of MeHg/fish/day (0.6 lg of MeHg/fish/day) compared to the control group (0.19 ± 0.03 lg·g^−1^ Hg dry weight). After 50 days, the Hg level rose to 46.2 ± 7.3 lg·g^−1^ dry weight, while the control group remained low (0.96 ± 0.08 lg·g^−1^ of Hg dry weight). These values denote MeHg’s high ability to penetrate the BBB. Furthermore, the transcription of genes involved in GABAergic and glutamatergic metabolism was determined after 25 and 50 days of exposure. No considerable changes were noted in the transcription of genes encoding the β subunit of GABA and glutamate receptors between the control and contaminated fish. However, after 50 days of exposure, a gene encoding the GABA degradation enzyme, GABA transaminase, revealed a 3.5-fold increase in transcription [87].

#### 3.2.2. Zebrafish as a Model for Heavy-Metal-Induced PD

PD is an advancing and chronic neurological condition and is mediated as the second neurodegenerative disease after AD. PD is expressed through the deficiency of dopaminergic neurons of the substantia nigra pars compacta (SNpc) [27,88]. There may be a harmful cycle in dopaminergic neurons, α-syn protein aggregation, and mitochondrial impairment [89]. The molecular pathways in PD pathogenesis remain uncertain; thus, scientists continuously research multiple mechanisms for neuronal loss, such as damaged protein quality pathways, deteriorated mitochondria, OS, nitrative stress, microglia stimulation, and inflammation. Brain analysis from post-mortem PD patients sustains these connections plus several more, including diminished GSH levels, autophagy dysregulation, proteasome deterioration, and α-syn aggregation. A combination of all these mechanisms can lead to OS, which has a compelling role in PD by destroying organelles and proteins, therefore, intensifying protein aggregation (α-syn), overcoming the body’s degradation systems and creating a never-ending loop for more OS [27].

Researchers have also observed markers of OS in the SNpc and dopaminergic neurons. There is an increase in OS levels in the brains of people with PD, regardless of whether the condition is or is not inherited. Another pathological feature of PD is the destruction of dopaminergic neurons in the substantia nigra. OS has a significant impact on the destruction of dopaminergic neurons in PD [89]. OS occurs when there is an imbalance between the body’s antioxidant responses and the generation of free radicals, such as superoxide dismutase (SOD), catalase (CAT), GSH, and glutathione S-transferase (GST) [59].

PD symptoms are linked to afflictions in the energy-producing structures within cells (mitochondria), imbalances in Ca and dopamine, and disruptions in autophagy and proteostasis. Mitochondrial dysfunction, dopaminergic metabolism, and neuroinflammation processes are the primary contributors to OS boost in PD. Changes in some proteins related to PD (protein deglycase, DJ-1; PTEN induced putative kinase I, PINK1; Parkin; α-syn; leucin-rich repeat kinase 2, LRRK2) may also contribute to this cycle, leading to mitochondrial dysfunction, resulting in increased ROS production and vulnerability to OS [89]. Inefficient oxidative phosphorylation can lead to the generation of ROS, resulting in mitochondrial dysfunction. Mitochondrial redox metabolism, proteolytic pathways, and phospholipid metabolism are potential sources of free radicals. While normal cellular signalling requires a low concentration of ROS, prolonged exposure to high levels of ROS can damage cellular macromolecules, such as DNA, proteins, and lipids, leading to necrosis and apoptotic cell death [90].

Alongside the neuronal damage, Lewy bodies’ occurrence in surviving neurons, linked with α-syn protein aggregates, is another trademark factor for PD [27]. Distinct cellular mechanisms, which are modified in PD pathophysiology, are connected to α-syn, being produced by the Synuclein Alpha (*SNCA*) gene. A diminution in α-syn decline can be generated genetically or not by breaking the lysosomal pathways. If the processing of α-syn is impaired or dysregulated, it can initiate ROS production and mitochondrial dysfunction. Furthermore, the malfunctioning mitochondria may influence α-syn in a feedback loop, even if it is not absolutely evident how this happens [91].

Among the main effects induced by Pb, along with other environmental pollutants and associated with PD, are mitochondrial dysfunction, changes in metal homeostasis, and protein aggregation, such as the primary component of Lewy bodies, α-syn [27]. Early-life exposure to Pb induces long-term toxicity in the CNS of zebrafish larvae, juveniles, and adults. The genes investigated are mainly involved in CNS development and neurotransmitter systems. Specifically, in exposure to zebrafish larvae, the messenger ribonucleic acid (mRNA) expression of *synapsin IIa* (*syn2a*) was upregulated. The inhibition of other neurodevelopment-related genes, namely *myelin basic protein a* (*mbpa*), *ELAV like neuron-specific RNA binding protein 3* (*elavl3*), *growth associated protein 43* (*gap43*), *glial fibrillary acidic protein* (*gfap*), and *v-fos FBJ murine osteosarcoma viral oncogene homolog Ab* (*fos*, a marker for increased neuronal activity), was significantly downregulated upon exposure to Pb in a dose-dependent manner. For genes related to the dopaminergic system, *catechol-O-methyltransferase a and b* (*comta* and *comtb*), expression was inhibited, while the expression of other genes, including *nuclear receptor subfamily 4, group A, member 2b* (*nr4a2b*), *dopamine beta-hydroxylase (dopamine beta-monooxygenase)* (*dbh*), *tyrosine hydroxylase* (*th*), and *dopamine receptor D1b* (*drd1b*), was all significantly induced by Pb exposure. In addition, for crucial genes, which are functional in GABAergic, AChergic, and Serotonergic systems, they observed a general enhancement of mRNA expression after exposure to environmental levels of Pb [92].

Hg exposure or ingestion is correlated with the particular symptoms and consequences of PD [93]. Studies have reported a critical association between PD patients and Hg blood levels [89]. Furthermore, Dantzig et al. [93] observed that detectable levels of Hg in the blood were six-times more common in patients with PD than in healthy patients [93]. Hg causes cognitive and motor impairments and neuron loss in model animals [89]. MeHg can accumulate in the zebrafish brain [94,95], and at higher doses (10–13 ppm), studies have implied alterations in proteins associated with gap junctions and oxidative phosphorylation, meaningful increases in *metallothionein 2* (*mt2*), and mitochondrial dysfunction [30,95]. According to Amlund et al. [95], low-level exposure of zebrafish to MeHg for 56 days or ten months can lead to changes in swimming behaviour, decreased foraging efficiency, and inhibition of membrane adenosine deaminase (ADA) at mid-range levels (between 720 ppb and 6 ppm) [95].

In addition, Cd has similar adverse effects to those encountered in Hg and Pb exposure. Cd contributes to an increase in ROS activity production, nitric oxide, and malondialdehyde (MDA) levels in the brain alterations of Cu/Zn-SOD and CAT levels, upregulating the mRNA, protein, and activity levels of inducible nitric oxide synthase (iNOS) and ciclooxigenase-2 (COX-2) in the brain (associated with inflammatory response) [96]. Furthermore, Cd increases the mortality rate in the zebrafish embryos, lowers the number of developing primary and secondary motor neurons, and amplifies neuromast damage in zebrafish embryos [97].

Scientists argue that neuroinflammation (microglia and astrocyte activation) processes are another pathological hallmark of PD. The CNS contains resident immune cells called microglia, which play an essential role in tissue repair and cellular homeostasis. Alterations in the inflammatory response can induce damage, safeguard against injury, or directly indicate neuronal deterioration. Contemporary scientific researchers maintain that inflammation has a key role in neurodegeneration. Therefore, PD patients’ brain analysis confirms the presence of inflammatory modifications, which, along with genetic changes in certain immune-function-related genes, pose a higher risk of PD development [98].

Exposure to heavy metals represents one of the most frequent occupational problems, and, as a result of exposure to heavy metals, dysfunctions in diverse organs can occur. For example, exposure to Cd is common among smelters and welders. The latest investigations indicate that Cd generates nerve cell death in embryos and adult fish [99,100]. Chronic exposure of zebrafish to a concentration of 1 mg·L^−1^ Cd for 16 days revealed effects on the neuroglial component from the first days of treatment [100].

Xu and his colleagues [101] explored the possible developmental neurotoxicity of Cd^2+^ at environmentally relevant levels in the context of microglia-mediated neuroinflammation in the connection between Cd^2+^ exposure and neurodegenerative diseases. The study gave unprecedented comprehension of the cellular and molecular mechanisms of Cd^2+^ toxicity in the zebrafish CNS. Furthermore, they investigated the implication of the wingless-related integration site (Wnt)/β-catenin signalling pathway in Cd-induced neurological disorders and neuroinflammation. The Wnt/β-catenin signalling pathway is involved in a range of biological processes comprising neurodevelopment, cell survival, and cell cycle regulation, as well as microglial, presenting one of the main targets of Cd^2+^ neurotoxicity. The results revealed that environmental Cd^2+^ exposure significantly affected the expression of major factors in the zebrafish Wnt/β-catenin signalling pathway [101].

### 3.3. Experimental Studies Linking Heavy Metal Exposure in Zebrafish and Neurological Disorders

Experimental studies revealed heavy metals obstruct various intracellular processes, therefore, being responsible for particular pathogenic mechanisms indicative of neurodegenerative diseases, such as dysfunction in the activity of mitochondria and protein, OS, and brain inflammation. Consequently, these aspects may generate impairment in the CNS functioning and changes in the behaviour and movement of fish [102]. The zebrafish behaviour is analysed to evaluate the neurotoxic effects of heavy metals, which decrease the mobility of the fish in water (swimming performance test), and could lead to memory loss (memory test) and a decline in the level of aggressive behaviour (mirror test) [103].

#### 3.3.1. Heavy Metals’ Effects on Zebrafish Swimming Performance Tests

Scientific experiments demonstrated that heavy metals [69,103], such as Cd, Hg, and Pb [69], affect swimming performance and cognitive processes in zebrafish and probably have toxic effects on the zebrafish’s CNS and behaviour [69,103] (Table 1), although supplementary analysis is required to explain this relationship [69]. However, these investigations have primarily focused on the toxic effects of heavy metals on zebrafish rather than specifically on their impact on swimming performance. Heavy metal exposure may be associated with neurological disease development related to AD and PD. Overall, it is not entirely clear how heavy metal exposure affects swimming performance in zebrafish [69].

Although swimming performance tests have been criticized in the past for lacking standards of measurement and guideline values, establishing clear protocols for representative species and proper baseline assessments of control groups should solve these issues. Behavioural endpoints that may be determined through swimming analysis include acceleration, frequency, turn angle, the time designated in different swimming modalities, vertical or horizontal distribution of individuals, and startle response. The neurotoxic effects of harmful substances could be assessed with the help of changes in posture or the pattern of swimming movements [107]. Behavioural abnormalities, such as disturbance parameters of swimming performance (total travelled distance, swimming speed, maximum acceleration, clockwise rotation, and activity or inactivity), were associated with the response to the effects of heavy metals, such as Hg, Pb, and Cd [103].

Contemporary scientists commonly use fish swimming performance testing because changes in predator–prey interactions are better observed, and effects on reproduction, habitat change, migration, and dispersal are distinguishable [108,109]. As observed in Sperandio’s [104] experiments, Cd-exposed fishes marked an increase in boldness but a decline in freezing behaviour compared with the control group. During the behavioural tests, exposed fish exhibited an impaired swimming ability and overloading swimming twists for a few minutes. In these conditions, Cd leads to deficiency in the fish functions that diminish the fitness of individuals, raises the mortality rate because of increased predatory risk, and ultimately affects the community through predator–prey relationships [104].

A group of scientific researchers led by Agnisola [110] ascertained that 0.3 and 3 mg·L^−1^ Cd chloride (CdCl_2_) administration to zebrafish adults for 15 to 30 days affects muscle fibres, thus, significantly affecting swimming performance. Results of muscle tissues’ cytological assays with haematoxylin–eosin-stained sections revealed Cd exposure led to myofibril disorganization, thinning in the muscle fibres, and stretching of the endomysial fibres [110]. Moreover, CdCl_2_ exposure for 72 h from early stages (6 h post fertilisation, hpf) in zebrafish induces a substantial reduction in the movement of individuals in the larval stage—78 hpf, affecting behavioural parameters due to changes in skeletal muscle fibre and glycogen. Using a DanioVision equipment system, the experts observed a decrease in swimming ability (speed, distance travelled, headings, and the number of collisions), directly proportional to the increase in the dose. The percentage of successful hatching in zebrafish larvae, which were treated with CdCl_2_ concentrations of 9, 18, 36, and 72 µM, substantially diminished in a dose-dependent manner, respectively, 83%, 76%, 54%, and 35%, compared with the control group [111].

Zebrafish larvae exposure to Pb registered reduced locomotor activity, whilst exposure to Cd disrupted behavioural rhythms, which were associated with the disruption of specific circadian genes. Researchers applied inductively coupled plasma mass spectrometry (ICP-MS) to estimate the internal exposure levels of Pb and Cd in zebrafish larvae (5 days post fertilisation—dpf) and a video-tracking system to observe the variations in behavioural rhythm. Further, quantitative polymerase chain reaction (qPCR) and Jonckheere-Terpstra-Kendall (JTK)-Cycle analysis were used to determine the changes in the expression of genes linked with melatonin-related molecules and clock genes. They determined the mean values of Pb and Cd, which were significantly higher in the groups exposed to Pb and Cd compared to the group exposed to a combination of Pb and Cd. Additionally, Cd and Pb administration to the model generated an antagonistic effect on the locomotor activity and the behavioural rhythm [112].

Tu et al. [113] exposed wild-type zebrafish embryos and larvae to Cd and Pb to establish their impact on locomotor activity and mortality, malformation, and bioaccumulation ratio. Treatment solutions of Cd (0.05, 0.1, 0.2 µM) and Pb (2.5, 5, 10 µM) generated high mortality, malformation, and bioaccumulation rates, which intensified along with an increase in dosage and duration of exposure of zebrafish embryos and larvae. Researchers monitored the zebrafish larvae’s swimming movement with a video tracking system and observed that developmental exposure to CdCl_2_ and Pb acetate (Pb(C_2_H_3_O_2_)_2_) caused a radical decrease in the swimming intervals and acceleration. Zebrafish embryos exposed to Cd exhibited a diminished swimming distance, which was notable, starting from a 0.1 μmol·L^−1^ CdCl_2_ concentration [113].

On the other hand, the continuous exposure of zebrafish embryos to Pb(C_2_H_3_O_2_)_2_ (0.05–0.7 mM) in an embryonic medium, starting from 0 to 6 days post hatch, had effects on the swimming distance, thus, detecting a causal relationship between the decrease in this distance and the increase in concentration [114].

An investigation led by Strungaru and colleagues [103] highlights that Hg exposure in zebrafish provokes changes in their behaviour, reduced swimming activity, and variations in response to stimuli [103]. Thus, researchers observed visible activity changes in the group exposed to a concentration of 15 µg·L^−1^ MeHg chloride (CH_3_HgCl) in terms of swimming performance starting 12 h after administration until the end of exposure (72 h). They noticed that acute exposure to a high CH_3_HgCl concentration affects the total distance travelled by the fish, ascertaining that this parameter decreases with exposure over time. Additionally, the swimming speed, maximum acceleration, and clockwise rotations decreased over time with exposure, particularly in the case of the group exposed to a higher CH_3_HgCl concentration, reaching no visible changes in these parameters in the group where 1 µg·L^−1^ CH_3_HgCl was exclusively administered [103]. In a similar manner to Strungaru et al. [103], Green and Planchart [69] found that Pb exposure generated shifts in the zebrafish behaviour, differences in their feedback to stimuli, and a reduction in their swimming movement [69].

#### 3.3.2. Use of the Social Test on Zebrafish to Assess the Implications of Heavy Metals in Neurological Disorders

The development of a wide range of tests to measure the cognition and behaviour of the zebrafish has prepared everything for its use in various studies conducted under optimal conditions. Animal models have certain limitations, and, to an extent, the zebrafish presents evident disadvantages, too. As a result, they incorporate particular challenges in quantifying their behaviour [12].

Social behaviour in zebrafish can be characterised as their preference to live near conspecifics. This type of behaviour determines the way one individual responds to a social stimulus [115]. Their social behaviour might be associated with their nature, being a collective species and living in banks composed of changeable numbers of individuals. They have authentic expressive conduct in the condition of fear and anxiety [115,116]. Shoaling, one of the most significant aspects of the behaviour recognised in zebrafish as their tendency to form compact groups, can be assessed and induced in a vast range of ways in the laboratory [3]. The social preference tests developed for zebrafish are similar to those used for spinners. The social tests are composed of two phases: the habituation phase, when the fish is left in the new environment to adjust and explore the environment, and the interaction phase, where a group of zebrafish will be added [115]. Social preference is measured by assessing the time spent by the fish close to the social stimulus area. This period is expressed as a percentage of the total time. The preference zone can be divided into a “strong” or a “weak” social preference zone, depending on the distance from the chamber where the social stimulus is [12].

Zebrafish are an excellent model for studying a broad range of diseases [115,116]. Furthermore, scientists began measuring other essential facets of social behaviour, such as the developmental trajectory and the neurobiological correlations. These tests provide valuable information regarding human brain illnesses connected with abnormal social behaviour. A significantly affected social behaviour is described by certain neurological diseases, such as autism spectrum disorder and schizophrenia, and distinct neuropsychiatric disorders, including depression and anxiety. Thus, the zebrafish model, which provides a vertebrate system’s biology with easily measured and generated social behaviour, may have relevant translational importance. Scientific reports propose that different aspects of nearly all brain conditions tested in rodent models can be examined in zebrafish in elementary approaches [3].

However, Thi and his scientific team [81] did not witness significant changes in shoaling and social interaction tests when zebrafish were exposed to a Pb concentration of 50 ppb for 30 days [81]. In contrast, Zhao et al. [117] discovered neurobehavioral abnormalities in larval zebrafish, including locomotor and social behaviours, predominantly induced at 20 μg·L^−1^ Pb concentration [85]. The Pb exposure experimental settings involved the zebrafish embryos’ (6 hpf) treatment with Pb concentrations of 5, 10, and 20 μg·L^−1^ for 144 hpf. During tests, scientists monitored the zebrafish models’ social behaviour, movement, number of turnings, and gene expression related to brain-derived neurotrophic factor (BDNF) signalling. Thus, zebrafish larvae exhibited hypoactivity in locomotion and turning behaviours during the dark period, contrasting with hyperactivity in a two-fish social test for the light period. The Pb concentrations registered in the larval samples were 4.54 ± 0.07, 7.11 ± 0.08, and 12.83 ± 0.27 μg·g^−1^ wet weight corresponding to the 5, 10, and 20 μg·L^−1^ Pb exposure groups. The research team witnessed a considerable downregulation of genes encoding BDNF and the N-methyl-D-aspartate glutamate receptor (NMDAR), indicating the implication of the NMDAR-dependent BDNF signalling pathway. Exposure to Pb at environmentally significant concentrations during developmental stages disrupts the NMDAR-dependent BDNF signalling, leading to neurobehavioral impairments in zebrafish larvae [117].

#### 3.3.3. Use of the Mirror Test on Zebrafish to Assess the Implications of Heavy Metals in Neurological Disorders

Looking in the mirror, we see a reflection of ourselves. Humans are conscious that it is merely a reflection, though a few animal species can distinguish that their reflection in the mirror is not another specimen that would pose a danger to them. Zebrafish fail the self-recognition test. Scientists can successfully use the mirror test to measure their aggression because zebrafish perceive the image reflected in the mirror as a rival [118]. In this case, a mirror test would be a good indicator of aggression, especially in fish. Nearly every known animal species, including fish, treat the mirror image as a specific aspect. Since each specimen is different, their responses may also be relatively different. Fish show this aggressive behaviour because the other fish “in the mirror” represents a threat to them. Although fish cannot recognise themselves in a mirror, they notice that their opponent is the same size and makes the same moves. Three different types of aggression can be identified: overt aggression, restrained aggression, and the time the fish spends close to the other individual. In addition, the time spent near the mirror, the number of approaches, bite attempts, mirror reflection pursuits, and the frequency of raising the dorsal, pectoral, anal, or caudal fins are used to outline the hostile attitude [119].

Aggressive behaviour is associated with territory defence, antagonistic interactions between group members, competition for food or mating, anti-predator behaviour, or prey capture. Some aggressive displays (fighting) are usually substituted with ritualising activities to establish dominance without injuring or killing their opponent. Changes that occur as a result of aggressive behaviour can damage or decrease the physical condition of the fish, increase the probability of losing a contest, or being injured after an inappropriate escalation of such a fight; thus, the chances of success in obtaining resources, territory, or food decrease [11]. Behavioural analysis is an excellent way to identify changes in the body. Expanding the area of application of behavioural tests is imperative. Throughout experiments studying neurobehavioral changes, both neurotic and typical, the zebrafish is an increasingly tested model [120]. Fontana et al. [121] analysed aggressive behaviour in zebrafish through the following parameters: duration, number, and average persistence of aggressive episodes combined with latency to attack the mirror. Motor and locomotor activities are also assessed through angular velocity and distance travelled, running automated analysis to perform the overall swimming pattern of zebrafish in these tests [121]. Recording aggressive elements without risking damaging the animal integrity is the main advantage of the mirror test [120].

A scientific team led by Gherlai et al. [118] and Kalueff et al. [122] investigated the exploration of zebrafish using parameters, such as the time spent in tank areas, transitions to basin sections, and the average duration of entry in those segments. When the fish presented combative behaviour through fin erection, the episodes were counted and associated with undulating body movements, biting towards the mirror, and fast swimming. Aggressive behaviour ends when the animal ceases to nibble at the opponent’s image [118,122]. A surge in the number and extent of attacks on the mirror is recognised as a higher aggression score. The lowest record is determined by the number of entries in the proximal sections of the tank that allow the fish to attack the competitor’s image. A higher hostility score is considered an increase in the amount and duration of the offensive behaviour towards the mirror. At the same time, a decreased score is expressed by absent aggressive displays, indicating exploration of the opponent’s image with less aggression [121].

Strungaru and his research colleagues [103] conducted a laboratory test with CH_3_HgCl zebrafish exposure at 1 μg·L^−1^, 15 μg·L^−1^, and a control group of 0 μg·L^−1^. The experiment associated behaviour assessments (swimming performance, mirror, and memory test) with the results obtained in those concentrations. SOD values significantly increased in the zebrafish group exposed to 15 μg·L^−1^ CH_3_HgCl. MDA activity also escalated in the lot exposed to 15 μg·L^−1^, although the MDA activeness did not increase substantially in the group exposed to 1 μg·L^−1^ CH_3_HgCl. Behavioural tests reveal that significant brain function damage occurs for nearly all variables during exposure, even at low concentrations. The outcome of those examinations highlights the effects of Hg in OS for SOD and MDA. The OS test evaluates the neurotoxicity of pollutants and, additionally, the physiological responses of organisms. Furthermore, the analysis assessed the reactions of the fish reflected in the mirror, a phenomenon known as the induced aggression test. The total distance travelled can be interpreted as one of the aggressive behaviour parameters. A short radius of migrating in the arm with the stimulus means a limited competitive attitude. The statistics confirm that CH_3_HgCl considerably affects the intensity of the hostile nature by diminishing it. Less aggressive specimens in ecosystems may have problems with predator avoidance, competition, and territorial protection [103].

A series of nine assays (mirror biting, social interaction, shoaling, predatory avoidance, short-term memory test, three-dimension locomotion, circadian rhythm locomotor activity, novel-tank exploration, and colour preference), implemented by a Taiwanese group of scientists, explored the behavioural impact of long-term heavy metal exposure on adult zebrafish [81]. According to Bui Thi et al. [81], chronic exposure to Pb induces anxiety, memory loss, and aggression in zebrafish. Zebrafish exposure to a low environmental concentration of Pb (50 ppb PbCl_2_, for one month) may produce an accentuated reduction in aggressive behaviour, which could impair memory and may lead to the diminution of their survival fitness [84]. Behavioural analysis results, especially from the mirror biting test, reveal that Pb inflates anxiety and stress levels through a lengthy freezing period and exploratory behaviour reduction. Further, the treated zebrafish models displayed insignificant aggressive behaviour, as demonstrated by a reduced duration on the mirror side. Nevertheless, the predator avoidance tests did not provide compelling transformations in zebrafish behaviour after chronic exposure to Pb. Assessment of the neurotransmitters and stress biomarkers from biochemical analysis of the cerebral tissue showed a high level of cortisol and reduced melatonin and serotonin levels in the brain, resulting in altered behaviour of exposed zebrafish [81].

Zheng et al. [123] carried out a study dealing with the anxiety-like neurobehavior of zebrafish caused by Pb exposure. Unlike the control group, the values in hatching rates were lower at Pb exposure concentrations of 12, 24, and 48 μmol·L^−1^. The zebrafish embryos’ mortality rate was higher than the control group. At the same dose, zebrafish larvae malformation rates were higher than in the group control. The turn angle parameter was higher than the control group parameter, yet, activity, thigmotaxis, and movement speed were significantly lower than the initial behaviour of the control group. MDA concentration and ROS levels in the brain were higher in zebrafish exposed to Pb, unlike the control group. On the other hand, noradrenaline and dopamine levels in the larvae brain were lower than in the control group. The presence of 5-hydroxytryptamine and corticotropin-releasing hormone, with stress response involvement, in the larvae brain exposed to Pb was superior to that in the control group [123].

#### 3.3.4. Correlation between Behavioural, Biochemical, Histological, and Bioaccumulation Experimental Studies Linking Heavy Metal Exposure in Zebrafish and Neurological Disorders

The contribution of zebrafish models to developing heavy metals’ neurotoxicity research is undeniably tremendous. An increasing number of experimental studies focus primarily on investigating the behavioural, toxicological, and neurological effects of heavy metals in embryos, larvae, and adult zebrafish. Recent assays inquired about accumulation and histopathological changes in zebrafish tissues, as well as physical or biochemical transformations in the brain (especially of the CNS) regarding neurodegenerative diseases, related to individuals exposed to particular heavy metal concentrations. Table 2 introduces experimental research performed exclusively on the zebrafish model, exposed to different concentrations of heavy metals. Accumulation properties of heavy metals highlight significant histopathological changes in zebrafish organs, such as the brain, liver, or skeletal muscle.

Innovative research reveals that zebrafish larvae exposed to MeHg present behaviour alterations, such as increased spontaneity and swimming activity, interfering with GR signalling [124].

According to Maximino and his colleagues [125], a 24 h exposure to MeHg (1000 or 5000 µg·L^−1^) decreases extracellular serotonin levels and increases the extracellular levels of tryptamine-4,5-dione, a partially oxidized metabolite of serotonin. The experiment showcases changes in an OS marker, which increased the formation of MDA. The OS outcome is mitochondrial dysfunction and originates from tryptamine-4,5-dione [125].

Exposure to HgCl_2_ in early developmental life results in decreased locomotor activity in zebrafish larvae. This has been proved through reduced spontaneous swimming speed and increased rest in response to aversive stimulation in open-field tests. HgCl_2_ affects glycogen metabolism, diminishes energy levels for the motor response, and generates permanent changes in neural patterns, including inhibition of AChE and disruption of glutamatergic and glycinergic neurotransmission, affecting swimming behaviour development [126].

Senger and his research team [127] conducted a chelation study to observe the inhibition caused by HgCl_2_ on soluble membrane ADA activity and gene expression in the zebrafish brain. At 24 h (acute) and 96 h (subchronic), ADA activity in soluble and membrane fractions decreased from exposure to 20 µg·L^−1^ HgCl_2_, leading to potential alterations in adenosine and inosine levels. Brain membrane subchronic exposure inhibited the enzyme. These data imply that HgCl_2_ alters extracellular and intracellular purine metabolism, potentially contributing to its neurotoxic effects [127].

Heavy metal neurotoxicity may be established through changes in the neurotransmission systems from zebrafish adult models [79]. Richetti et al. [79] correlated Hg and Pb exposure to a pollutant decrease in AChE activity in zebrafish. After exposure to HgCl_2_ for 24 h, the researchers observed a significant decline in the antioxidant capacity of the exposed zebrafish against peroxyl radicals, contrasting with results from the control group (*p* < 0.05). These findings suggest that heavy metal exposure may diminish signal transmission in the zebrafish brain tissue through transformations in cholinergic transmission and antioxidant competence, among other effects. Contrary to these findings, Cd generated no alterations in AChE activity or antioxidant capacity [79].

The exposure of zebrafish to HgCl_2_ or Pb(C_2_H_3_O_2_)_2_ had different effects on nucleotide hydrolysis. HgCl_2_ caused the inhibition of ATP, ADP, and AMP hydrolysis after 96 h, but after 30 days, ATP hydrolysis returned to normal, and ADP hydrolysis increased. Pb(C_2_H_3_O_2_)_2_ reduced ATP hydrolysis after 96 h and inhibited ATP, ADP, and AMP hydrolysis after 30 days. Real time polymerase chain reaction (RT-PCR) analysis indicated no change in nucleoside triphosphate diphosphohydrolase 1 (NTPDase1) and ecto-5′-nucleotidase expression after 30 days of exposure to both metals. Hg^2+^ and Pb^2+^ can impact ecto-nucleotides activities, a significant pathway in purinergic signalling control [128].

Both HgCl_2_ and CH_3_HgCl may increase the number of apoptotic cells in the brain, downregulate the expression of *shha*, *neurogenin 1* (*neurog1*), and *neuronal differentiation 1* (*neurod1*) genes, and reduces neurotransmitter levels (tyrosine, dopamine, and tryptophan). The neurotoxic mechanism of HgCl_2_ and CH_3_HgCl was analysed by examining the expression of these three genes involved in early neural development. The *shha* gene controls crucial developmental processes, such as neural tube patterns, neural stem cell proliferation, and survival of neurons and glial cells. Neurogenin is a marker for neuronal precursors regulated by *shha* and *neurog1*, expressed in the neural plate, which can induce *neurod1* expression, causing proliferative neural precursor cells to become post-mitotic neurons. In addition to gene downregulation, both substances decrease body length and eye size and affect hatching time, but only CH_3_HgCl significantly declines the average moving distance and trajectory. HgCl_2_ disturbed trajectory still was not observed, nor any significant decrease in moving distance [129].

Gonzalez et al. [130] established that the liver was the only organ that underwent a demethylation action by measuring the total Hg and MeHg concentrations at all three organ levels. Although, after 63 days, the cerebral tissue cumulated an elevated level of Hg concentration of 63.5 ± 4.4 µg·g^−1^, there were no visible modifications in the expression of thirteen genes implicated in various functions (such as apoptosis, metal chelation, DNA repair, mitochondrial metabolism, antioxidant defence, and organic compound efflux). Gene expressions related to mitochondrial metabolism and ROS production (*cytochrome C oxidase subunit I, coxI*; *cytoplasmic and mitochondrial superoxide dismutases*, *sod* and *sodmt*) started in the skeletal muscle and liver, therefore, illustrating an impact on these mechanisms. However, the study revealed that skeletal muscle is not only a critical storage reservoir for MeHg but is also affected by MeHg contamination. Liver and skeletal muscle marked contrasting patterns of gene expression for *mt2* and *RAD51 recombinase* (*rad51*) genes [130].

Pb is considered a highly toxic substance with multiple effects on the organism, increasing ROS production and the mRNA expression levels of antioxidant-relevant genes and inducing OS [131,133,136]. Auxiliary effects are spinal malformation (curvature) [132] and a low survival rate [131]. In zebrafish embryos, different concentrations of Pb and Cd may decrease the survival rate, hatching rate, and embryo activity [131], which leads to apoptosis. Likewise, exposure of zebrafish to Pb(C_2_H_3_O_2_)_2_*3H_2_O for 30 min enhances hyperactivity (increases significant distance covered, swimming speed, and speed under illumination) [136]. Pb favoured oedema formation in the cerebral area, defects in swim bladder inflation, and a decline in hatching time and body length on zebrafish larvae [133].

According to Dou and Zhang [114], embryos and larvae exposure to Pb(C_2_H_3_O_2_)_2_ may induce behavioural alteration as sluggish action, slow swimming movements, and slow escape action. Other alterations in the organism may include an increased mortality rate and incidence of “S” body malformation, downregulation expression of two genes in the diencephalon and dorsal hindbrain, extensive apoptosis of neuron cells and downregulation expression of two genes related to a decrease in the number of neural cells [114].

Zhang and his colleagues [67] tested embryonic zebrafish brains using anti-acetylated alpha-tubulin staining, through a series of eight exposure intervals during 36 hpf, to detect if transformations in neuronal growth and transport function of axons are the source of developmental Pb exposure. Reductions in axonal density for distinct axon tracts in the brain of embryonic zebrafish exposed to Pb treatment correlated with the downregulation of *shha*, *epha4b*, and *netrin2* genes. Overall, a comparison of the Pb concentration in the control group with the Pb-treated group at each time point revealed a significant increase in Pb accumulation in the tissue of the Pb-treated group (*p* < 0.0001). The measured mean values of Pb concentrations at all time points (≥404.39 ± 57.49 ng·g^−1^) were significantly higher than those in the initial treatment solution (65.64 ± 1.45 ppb). These statistical values for Pb from environmental exposure prove the bioaccumulation of Pb in the zebrafish tissues. Scientists separated the 24 hpf Pb-treated group of zebrafish embryos in fish bodies and chorion to determine Pb concentrations and discovered a significantly higher concentration of Pb in the bodies (184.94 ± 26.30 ng·g^−1^) compared to the chorion (32.93 ± 8.09 ng·g^−1^) [67].

According to Wang et al. [92], increased Pb levels were associated with decreased larval moving distance, elevated acceleration in juvenile zebrafish, and increased travelling distance and velocity in adults. Other alterations in the organism are declining in survival rate, body length, slightly increased hatching rate, severe malformations (scoliosis, tail and head deformities), the tendency for neuron loss, abnormal neuronal varicosities, expressing a negative impact on neural development function, as it is a symptom of nerve damage and neurodegenerative diseases [92].

Moreover, exposure of zebrafish to Pb(C_2_H_3_O_2_)_2_*3H_2_O altered spontaneous movement, revealing hyperactivity in larvae (increased swimming activity and speed), and impaired learning/memory performance [134]. A team of scientists led by Kataba [135] found that *uncoupling protein 2* (*ucp-2*) and *B-cell lymphoma 2* (*bcl-2*) gene expression increased significantly in the exposed group. The upregulation of *coxI* and *tumour protein p53* (*tp53*) genes reveals an increase in ROS production, which led to increased expression of *ucp-2* and *bcl-2* as an initial protective mechanism. The u*cp-2* gene reduces ROS production during mitochondrial transport, and *bcl-2* counteracts the effect of elevated *tp53* expression and is an indicator of increased ROS. The significant increase in *coxI* expression in acutely exposed zebrafish larvae highlights the need for cellular responses to the ROS generated from exposure [135].

Zhu et al. [137] investigated the effects of Pb and BDE-209 on neurodevelopment in zebrafish larvae. Results emphasized HgCl_2_ increment malformation percentage and decreased dorsal axonal length. Data confirmed that this mixture had a synergic effect on this process, possibly because of the enhanced generation of ROS, lipid peroxidation (increased MDA levels), DNA damage, or perturbation of the antioxidant system (decrease in GSH, reduction in SOD activity). Further, in the co-exposed group to Pb and BDE-209, growth of the secondary motoneuron axons and inhibition of genes related to CNS development (*mbp*, *α1-tubulin*, and *gfap*) were observed [137]. In their article, Green et al. [138] disclosed that zebrafish embryos/larvae exposed to CdCl_2_ have behavioural alterations, such as hyperactivity and increased rotational movement. Other effects are reduction in otolith saccule diameter and pronounced changes in the otolith ultrastructure and expression level of an essential gene for vestibular Ca sensing and transport [138].

Zhang et al. [139] discovered that exposing zebrafish embryos to Cd resulted in induced malformations (reduced body size, trunk abnormalities, including hyphosis typified by dorsal convex curvature of the vertebral column and lordosis typified by dorsal concave curvature of the vertebral column, hypopigmentation, head hypoplasia, microphthalmia, reduced interorbital distance), increased mortality, reduction in expression of genes labelling the neural crest and its derivative pigment cells, blockage of neural crest formation, and inhibition of specification of pigment cells [139].

In addition, Ruiter and his scientific team [140] determined that Cd affects behaviour through impairment in stress handling (increased duration in stress response) and hyperactivity. It may also reduce embryo survival, alter DNA methylation, and alter body antioxidant physiology (decreased GPx, decreased GSH, increase GSSG, no effects observed on the activity of GR and SOD) [140].

Cd accumulated at a stimulated rate in the zebrafish body immediately after exposure, and researchers discovered an association between its ecotoxicity and relevant biochemical processes in the muscles and brain. Thus, after a 96 h Cd exposure, SOD, CAT, and AChE levels of activity recorded a significant enhancement. Yet, they noted behavioural alterations, such as decreased activity, loss of equilibrium, restlessness, abnormal swimming behaviour, and rapid gill movement [141].

Three zebrafish exposure groups (C1, C2, and C3) registered, in the tissues from the whole fish body, bioconcentration factors of Cd 14.3, 18.16, and 13.48 mg·L^−1^. However, a decrease in Cd bioaccumulation in the fish body was detected in C3 when diverting from 15 to 25 days of exposure, with the Cd concentration in the treated fish presenting an ascending trend and reaching peak values of 95.33 (C1), 66.04 (C2), and 24.42 (C3) µg·g^−1^ at day 25. Histopathological examination of skeletal muscle and brain tissue from Cd-exposed zebrafish for 5 and 25 days uncovered abnormalities that were aggravated along with the duration of exposure (degeneration of Purkinje cells, different extents of granule cell loss, neutrophil loss, aggregation area of gliosis, various degrees of necrosis). Cd induces behavioural alterations, including erratic swimming, aggressiveness, and hyperventilation [142].

## 4. Conclusions

Heavy metals (Cd, Hg, and Pb) raise significant concern regarding the CNS and BBB functions, induce neuronal cell cytotoxicity, and are involved in ROS production, mitochondrial dysfunction, and apoptosis.

Zebrafish is an appealing model with enormous potential in various research fields, considering its multiple advantages (it has a human-like nervous system, a short life cycle, and reproduces quickly and in abundance). It has proven its importance through the results obtained and has the potential to be a front-list model. Heavy metal exposure can have a negative impact on the zebrafish’s health, such as swimming difficulty, behavioural changes, and abnormal development. These are used to model and understand how heavy metals affect human organisms. Scientific experiments on zebrafish have provided insights into how heavy metals affect the nervous, reproductive, and immune systems. Furthermore, they contributed to the identification of possible treatments for heavy metal exposure.

The results suggest that Cd, Hg, and Pb exposure is involved in CNS neurotoxicity, which may lead to neurodegenerative diseases (AD and PD). These heavy metals can interfere with the normal functioning of the nervous system and cause damage to neurons. For example, we have observed that exposure to Pb can lead to reduced connections between neurons and cause cognitive decline. There is also evidence that Hg exposure can cause neuron degeneration and contribute to cognitive deterioration. Overall, zebrafish represent a valuable model in studies of heavy metals and can help improve the understanding of these substances’ effects on human health.

This review endeavoured to shed light on the profound neurotoxic effects of heavy metals, which are not limited only to exposed generation and are of tremendous concern to public health. Imperative investigation into specific genetic and molecular mechanisms of neurodegenerative diseases is still required. We must admit the limitation of zebrafish use in modelling the progression of neurodegenerative diseases as they are associated with ageing.

Although contemporary science is steadily advancing, neurodegenerative diseases invariably resemble a puzzle, with a few mixed or missing pieces. The mechanisms of how heavy metals induce neurodegenerative diseases are not fully understood. Moreover, different types of analysis may provide essential evidence that suggests the importance of environmental risk factors for neurodegenerative diseases. The study of brain tissue damage and other relevant molecular analysis should be added to behavioural ones, as behavioural tests alone are not enough to establish zebrafish as a useful model. Novel scientific papers are required to clarify the mechanisms of how heavy metals induce neurotoxicity and provide essential evidence proposing heavy metals as a critical environmental risk factor for neurodegenerative diseases.

## Figures and Tables

**Figure 1 ijms-24-03428-f001:**
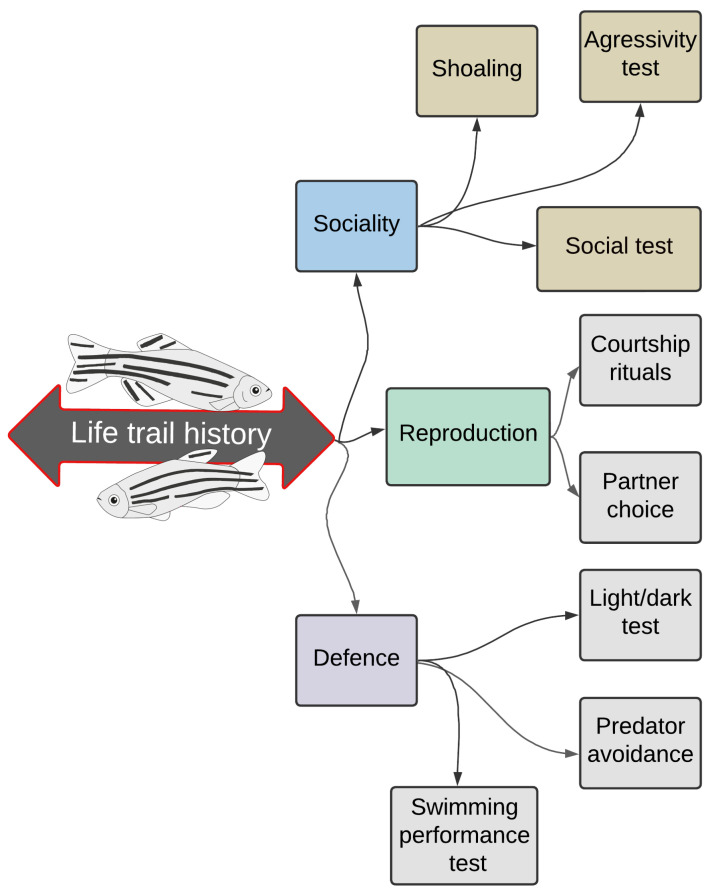
Laboratory experiments with different tests can provide essential information about toxicants’ impact on one or more behavioural domains in zebrafish (sociality, reproduction, defence), which are likely to affect these complex responses. The particular facts are related to the responses received at the population, species, community, and ecosystem levels (data modified from Dutra Costa et al. [11]).

**Figure 2 ijms-24-03428-f002:**
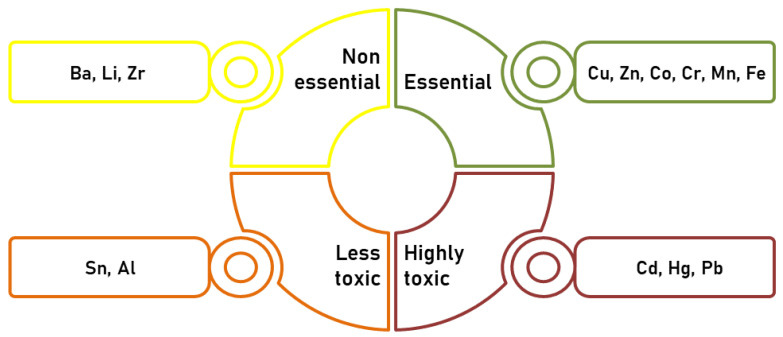
Heavy metal classification according to health importance (data modified from Raikwar et al. [20]).

**Figure 3 ijms-24-03428-f003:**
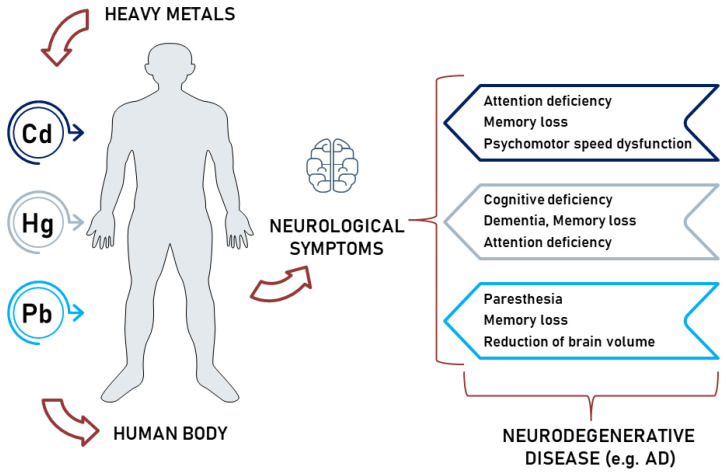
Exposure to highly toxic metals (Cd, Hg, Pb) induces neurological deficiency in the human body (data modified from Lee et al. [28]).

**Figure 4 ijms-24-03428-f004:**
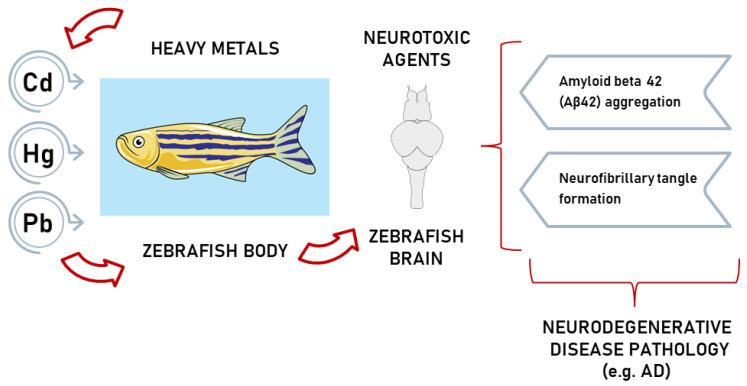
Exposure to highly toxic metals (Cd, Hg, Pb) induces neurological deficiency in the zebrafish body (data modified from Thawkar and Kaur [72]).

**Table 1 ijms-24-03428-t001:** Comparison between the heavy metals’ effects on human and fish behaviour and CNS.

Heavy Metal	Effects on Human Behaviour and CNS	Effects on Fish Behaviour and CNS	References
Cd	Causes muscle weakness, withdrawal, and impairs cognitive function, may be linked to the development of PD	Impairs swimming ability and causes deficiency in fish functions	[44,104]
Hg	Causes memory loss, other cognitive impairments and tremors	Reduces swimming activity and changes in response to stimuli	[103,105]
Pb	Causes deficits in cognitive functions and could be related to the development of AD	Impairs memory and learning, decreases locomotion (speed) and changes in response to stimuli	[69,106]

**Table 2 ijms-24-03428-t002:** Summary of zebrafish studies measuring behavioural, toxicological, and neurological effects of Cd, Hg, and Pb exposure.

Chemical	Developmental Stage	Exposure Duration	Behavioural Alterations	Other Alterations in the Organisms	Reference
MeHg	larvae	2 hpf–5 dpf	Increased spontaneity and swimming activity	Interfered with glucocorticoid receptor (GR) signalling	[124]
MeHg	adult	24 h	Presence of anxiety-like behaviour	Intensified increase in the amount of MDA, decreased extracellular serotonin, increased OS, produced mitochondrial dysfunction	[125]
HgCl_2_	embryos/ larvae	5–72 hpf	Motor deficit, disruption in the anxiety-like behaviour, and decreased swimming activity	Increased embryo mortality rate, biochemical changes in proteins, lipids, carbohydrates and nucleic acid	[126]
HgCl_2_	adult	24–96 h	No changes in swimming behaviour	Inhibited ADA activity	[127]
HgCl_2_	adult	24 h, 96 h, 30 days	Behavioural observations not noted	Altered AChE activity and antioxidant capacity	[79]
HgCl_2_	adult	24 h, 96 h, 30 days	No changes in the swimming pattern were observed	No observed mortality; during 96 h—inhibited the hydrolysis of adenosine triphosphate (ATP), adenosine diphosphate (ADP) and adenosine monophosphate (AMP); after 30 days—ATP hydrolysis returned to the control levels, ADP hydrolysis was strongly increased and AMP hydrolysis remained inhibited.	[128]
HgCl_2_	embryos (6 hpf)/ larvae	24 h	Trajectory disturbance, no significant decrease in the average moving distance	Delayed hatching, decreased body length and eye size, tail bending, increased number of apoptotic cells in the brain, downregulated neural development related genes, reduced levels of neurotransmitters (tyrosine, dopamine, and tryptophan)	[129]
CH_3_HgCl			Significant decrease in the average moving distance and trajectory disturbance		
CH_3_HgCl	adult (male)	7, 21, and 63 days	No decrease in motility	Generated higher levels of bioaccumulation in brain tissue, no increase in mortality	[130]
Pb Nitrate—Pb(NO_3_)_2_	embryos/ larvae	24–96 hpf	Behavioural observations not noted	Decreased survival rate in embryos, reduced hatching rate, increased ROS production and the mRNA expression levels of antioxidant-relevant genes	[131]
Pb(NO_3_)_2_	embryos (3–4 hpf)	1–7 dpf	Behavioural observations not noted	Decreased survival rate, increased apoptosis and transcriptional levels of two genes related to antioxidant defence and two apoptosis-related genes, spinal malformations (curvature)	[132]
Pb(C_2_H_3_O_2_)_2_	embryos/ larvae	0–6 dpf	Sluggish action, slow swimming movements, slow escape action	Increased mortality rate and incidence of “S” body malformation, downregulated expression of two genes in the diencephalon and dorsal hindbrain—extensive apoptosis of neuron cells, downregulated expression of two genes related to a decrease in the number of neural cells	[114]
Pb(C_2_H_3_O_2_)_2_	embryos (6 hpf)/ larvae	6–72 hpf	Behavioural observations not noted	No difference in viability, shortened hatching period and body length, defects of swim bladder inflation, oedema formation in cerebral area, hatching period and body length, increased ROS levels and the expression levels of OS response-related genes	[133]
Pb(C_2_H_3_O_2_)_2_	embryos (2 hpf)	12–36 hpf	Behavioural observations not noted	Alterations in neuronal growth, decreased axonal density, interfered with the expression of 3 genes involved in axonogenesis (downregulated *sonic hedgehog signalling molecule a* (*shha*) and *ephrin type-A receptor A4b* (*epha4b*) genes, and overexpression of *netrin* (*netrin2*) gene)	[67]
Pb(C_2_H_3_O_2_)_2_	embryos (2 hpf)/ larvae/ adults (exposure only during embryogenesis)	2–120 hpf	Decreased larval moving distance and acceleration, elevated acceleration in juvenile zebrafish, increased travelling distance and velocity in adults	Decreased survival rate, slightly increased hatching rate, decreased body length, severe malformations (scoliosis, tail and head deformities), tendency to neuron loss, extensive apoptosis, induced varicosities formation in adult zebrafish brain	[92]
Pb(C_2_H_3_O_2_)_2_	adult (exposure only during embryogenesis)	1–72 hpf	Behavioural observations not noted	Alterations in genes associated with nervous system’s development and function, more pronounced in a set of 89 genes associated with AD (including human homologous *APP* and *APOE*)	[64]
Pb(C_2_H_3_O_2_)_2_	adult	24 h, 96 h, 30 days	Behavioural observations not noted	Altered AChE activity but not antioxidant capacity	[79]
Pb(C_2_H_3_O_2_)_2_	adult	24 h, 96 h, 30 days	No changes in the swimming pattern were observed	No observed mortality; during 96 h—caused a significant decrease only on ATP hydrolysis; after 30 days— promoted the inhibition of ATP, ADP and AMP hydrolysis	[128]
Pb acetate trihydrate— Pb(C_2_H_3_O_2_)_2_*3(H_2_O)	embryos/ larvae	6–120 hpf	Altered spontaneous movement (decreased tail bend frequency in embryos), hyperactivity in larvae (increased swimming activity and speed), impaired learning/memory performance (reduction in the accuracy rate and an increase of time to reach the food end)	Induced malformations (bent spine)	[134]
Pb(C_2_H_3_O_2_)_2_*3(H_2_O)	embryos (2.5 hpf)/ larvae	96 hpf	Induced muscular twitching	Attenuated burst activity in embryos, induced changes in the mRNA expression levels of antioxidant and OS response enzymes	[135]
Pb(C_2_H_3_O_2_)_2_*3(H_2_O)	larvae (120 hpf)	30 min	Enhanced hyperactivity (significant increases in distance covered, swimming speed and mobile frequency as well as increased distance covered and speed under light illumination)	Upregulated the mRNA expression of genes related to increased ROS	[136]
Pb(C_2_H_3_O_2_)_2_*3(H_2_O) and deca-brominated diphenyl ether (BDE-209)	embryos (2 hpf)/ larvae	2–144 hpf	Reduced average swimming speed	No observed interference with hatching percentage, growth rate, or survival percentage, increased malformation (axial spinal curvature), decreased dorsal axon length, Downregulated the expression of three CNS genes (*mbp*, *α1-tubulin*, and *gfap*), increase in ROS, lipid peroxidation (increased MDA levels), DNA damage, perturbation of the antioxidant system, (decrease in GSH, reduction of SOD activity)	[137]
CdCl_2_	embryos/ larvae	4 hpf–7 dpf	Hyperactivity and increased rotational movement	Induced reduction in saccule otolith diameter, pronounced changes in the otolith’s ultrastructure, altered expression level of a gene important for vestibular Ca sensing and transport	[138]
CdCl_2_	embryos/ larvae	2.5–96 hpf	Behavioural observations not noted	Induced malformations (reduced body size, trunk abnormalities including hyphosis typified by dorsal convex curvature of the vertebral column and lordosis typified by dorsal concave curvature of the vertebral column, hypopigmentation, head hypoplasia, microphthalmia, reduced interorbital distance), increased mortality, reduced expression of genes labelling the neural crest and its derivative pigment cells, blockage of neural crest formation, inhibition of specification of pigment cells	[139]
CdCl_2_	embryos (≤1 hpf)	24–72 hpf	Affected the escape response	Increased mortality rate, no observed effects on hatching rate, lowered number of normally developing primary and secondary motor neurons, neuromast damage	[97]
CdCl_2_	embryos/ adult (exposure only during embryogenesis)	0–72 hpf	Hyperactivity, impaired stress handling (increased duration in the stress response)	Reduced embryo survival, altered DNA methylation, altered body antioxidant physiology (decreased glutathione peroxidase GPx, decreased GSH, increase oxidized glutathione GSSG, no effects observed on activity of glutathione reductase GR and SOD)	[140]
CdCl_2_	adult	0–96 h	Decreased activity, loss of equilibrium, restlessness, abnormal swimming behaviour, rapid gill movement	Increased AChE activity in the brain, increased SOD and CAT activity in the brain, decreased protein content in brain tissue, increased total lipid content, bioaccumulation factor in the body was 30 mg·L^−1^	[141]
Cd acetate—C_4_H_6_CdO_4_	embryos/ larvae	24–96 hpf	Behavioural observations not noted	Decreased survival rate in embryos, reduced hatching rate, increased ROS production and the mRNA expression levels of antioxidant-relevant genes	[131]
C_4_H_6_CdO_4_	adult	24 h, 96 h, 30 days	Behavioural observations not noted	Generated no alterations in AChE activity or antioxidant capacity	[79]
Cd	adult (female)	24 h, 96 h	Behavioural observations not noted	Increased ROS, nitric oxide and MDA levels in the brain, no significant alteration of Cu/Zn-SOD and CAT levels, up-regulated the mRNA, protein and activity levels of iNOS and COX-2 in the brain (associated with inflammatory response)	[96]
Cd	adult	0–25 days	Induced erratic swimming and aggressiveness, hyperventilation	Increased mortality rate, increased bioaccumulation factor values in body, histopathological changes in the brain (degeneration of Purkinje cells, different extents of granule cell loss, neutrophil loss, aggregation area of gliosis, various degrees of necrosis)	[142]

## Data Availability

The datasets used and analysed during the current study are available from the corresponding author upon request.

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
