# Peer review of "Zebrafish as a Potential Model for Neurodegenerative Diseases: A Focus on Toxic Metals Implications"

_ijms, 2023, doi:10.3390/ijms24043428_

Round 1
Reviewer 1 Report
This is a very valuable and well-prepared review of the literature on the use of Danio rerio to study the effect of heavy metals on human neurodegenerative diseases. I am full of appreciation for the authors who managed to select from thousands of papers those presenting experiments on Danio rerio with the use of the three metals Hg, Cd and Pb. They prepared the content of the article in a very professional way; they first provided general information about the neurotoxic effects of the three heavy metals discussed, then presented the possibilities of using Danio rerio for the study of human neurodegenerative diseases, and finally focused on the effects of testing zebrafish in this area. The article will be very helpful for young medical adepts looking for a remedy for still poorly understood neurodegenerative diseases and will probably have many citations.
I have only a few minor remarks that authors should take into account before publishing the article:
1) I found no mention of Minamata disease in the article. In my opinion, this is one of the better studied neurological diseases caused by methyl mercury, so it should be mentioned in the paper (e.g. in chapter 2.1.2 or line 262 of chapter 2.3.).
2) Usually, the purpose of the article is stated at the end of the introduction. This time the authors say there are "several aspects that require our attention", but they do not say what those aspects are. The reader may assume that these are the neurophysiological changes mentioned above in line 67. Instead, the point is that the zebrafish is a valuable model in the study of heavy metals and can help to better understand the effects of these substances on human health. This should be articulated at the end of the introduction.
3) Fig. 2 has 3 labels on the right that have a common brace indicating that these are neurological symptoms. These labels repeat information about symptoms, such as memory loss and attention deficiency. This may seem like unnecessary repetition. It would be good to start each label with a different metal, i.e. Cd, Hg, Pb, to avoid confusion.
4) There is an awkward note on line 550 which can be understood to mean that zebrafish and humans have a similar life cycle. You should add a comma and the word “short” instead of “and” before life cycle.
Author Response
Reviwer # 1. This is a very valuable and well-prepared review of the literature on the use of Danio rerio to study the effect of heavy metals on human neurodegenerative diseases. I am full of appreciation for the authors who managed to select from thousands of papers those presenting experiments on Danio rerio with the use of the three metals Hg, Cd and Pb. They prepared the content of the article in a very professional way; they first provided general information about the neurotoxic effects of the three heavy metals discussed, then presented the possibilities of using Danio rerio for the study of human neurodegenerative diseases, and finally focused on the effects of testing zebrafish in this area. The article will be very helpful for young medical adepts looking for a remedy for still poorly understood neurodegenerative diseases and will probably have many citations.
I have only a few minor remarks that authors should take into account before publishing the article:
- I found no mention of Minamata disease in the article. In my opinion, this is one of the better studied neurological diseases caused by methyl mercury, so it should be mentioned in the paper (e.g.in chapter 2.1.2 or line 262 of chapter 2.3.).
- We agree with this remark and we added a short description of the Minamata disease. Thank you for you input.
- Usually, the purpose of the article is stated at the end of the introduction. This time the authors say there are "several aspects that require our attention", but they do not say what those aspects are. The reader may assume that these are the neurophysiological changes mentioned above in line 67. Instead, the point is that the zebrafish is a valuable model in the study of heavy metals and can help to better understand the effects of these substances on human health. This should be articulated at the end of the introduction.
- We used a new paragraph at the end of the introduction for reiterating the purpose of the article. Thank you for you input.
- 2 has 3 labels on the right that have a common brace indicating that these are neurological symptoms. These labels repeat information about symptoms, such as memory loss and attention deficiency. This may seem like unnecessary repetition. It would be good to start each label with a different metal, i.e. Cd, Hg, Pb, to avoid confusion.
- We modified the colour of the contours from figure 2 to better emphasize the connection between the chemical symbols of the metals on the left and the list of symptoms on the right. Also, we changed the order of the second set of symptoms to avoid repetition. Thank you for your input.
- There is an awkward note on line 550 which can be understood to mean that zebrafish and humans have a similar life cycle. You should add a comma and the word “short” instead of “and” before life cycle.
- We included the suggested modification. Thank you for your input.
Reviewer 2 Report
This review aims to analyse and discuss the value of translational animal models used in neurological conditions such as Alzheimer’s disease (AD), Parkinson’s disease (PD), and epilepsy, particularly the benefits of animal models, therapies in patients, and what limitations exist. Present review has been performed analysis only on the English-written articles. It is generally well written with appropriate data analyses and an interesting discussion.
Author Response
Reviewer # 2.
This review aims to analyse and discuss the value of translational animal models used in neurological conditions such as Alzheimer’s disease (AD), Parkinson’s disease (PD), and epilepsy, particularly the benefits of animal models, therapies in patients, and what limitations exist. Present review has been performed analysis only on the English-written articles. It is generally well written with appropriate data analyses and an interesting discussion.
- Thank you. We appreciate your input.
Reviewer 3 Report
This review article is well written and its topic is one of importance to the field. However, there are several problems present in this current manuscript
- The relationship between the neurological diseases caused by heavy metal exposure and the different parameter tests (swimming speed and the mirror test) is not well established
-A brief addition of methodology used in the parameter testing papers should be made. How the tests were conducted and the controls used should be elaborated on.
-No clear convincing link is established and elaborated on for the use of Zebrafish as a model for the different neurological diseases as well as how effective they are as models for human CNS diseases. The Zebrafish are stated to have been used as models in several papers but no elaboration is made to show how this is done or how effective their use is.
-No mention of taking blood or tissue samples from the zebra fish to establish the levels of heavy metal accumulation as well as the physical changes to the CNS caused by exposure
-Main scope and focus of the paper can be further refined and made more concise.
Author Response
Reviewer # 3.
This review article is well written and its topic is one of importance to the field. However, there are several problems present in this current manuscript:
- The relationship between the neurological diseases caused by heavy metal exposure and the different parameter tests (swimming speed and the mirror test) is not well established
- We were considering adding the details mentioned above before the submission of the article but were short on time. Now we proceeded with inserting them in the revised version of the article. Thank you for your input.
-A brief addition of methodology used in the parameter testing papers should be made. How the tests were conducted and the controls used should be elaborated on.
- We agree with this remark and we added a few paragraphs with more details regarding the methodology of the parameter testing. Thank you for you input.
-No clear convincing link is established and elaborated on for the use of Zebrafish as a model for the different neurological diseases as well as how effective they are as models for human CNS diseases. The Zebrafish are stated to have been used as models in several papers but no elaboration is made to show how this is done or how effective their use is.
- We presented data regarding the similarities between the zebrafish and human brain. Also, we added a few remarks about the effectiveness of zebrafish as models in the last paragraphs from subchapter 3.3. Testing zebrafish as a research model for different neurological disorders. Thank you for you input.
-No mention of taking blood or tissue samples from the zebra fish to establish the levels of heavy metal accumulation as well as the physical changes to the CNS caused by exposure.
- We agree with this remark and we added a new subchapter (4. Heavy metals’ bioaccumulation and histopathological changes in zebrafish tissues) for clarifying the connection between levels of heavy metals in the environment and in tissues of zebrafish and the changes in cerebral activity. Thank you for you input.
-Main scope and focus of the paper can be further refined and made more concise.
- We used a new paragraph at the end of the introduction for reiterating the purpose of the article. Thank you for you input.
Round 2
Reviewer 3 Report
Your recent additions and adjustments based on the comments made by the reviewers on the manuscript are appreciated however, the title should be narrowed down to zebrafish as the majority of the manuscript discuss the uses of zebrafish as a model for neurodegenerative diseases and scarcely are other aquatic organisms mentioned.
Furthermore, the manuscript in its current form is not accepted for publication.
I suggest a restructuring of the manuscript to the following to better fit your newly defined scope:
An Introduction that includes the relevant common structures and conserved mechanisms in the brain between humans and aquatic animals such as zebra fish in the context of modeling human neurodegenerative diseases.
Establishing the link between heavy metal exposure and the neurodegenerative diseases.
Explaining the relevant markers for each disease that exist for both humans and zebra fish (that are affected by exposure to heavy metals)
Using the mentioned relevant markers to establish a way of using the aquatic organisms as a model for each disease (i.e. an experiment that exposes zebrafish to heavy metals and measures the accumulation of the mutated tau protein associated with dementia in the zebra fish CNS or other specific brain damage that translates to the damage caused by a neurodegenerative disease in the human CNS )
Sections that describe the experiments carried out on zebra fish including their relevant methodology and tissue sampling (for heavy metal exposure to areas relevant to the CNS function)
A conclusion that includes the limitations of the aquatic models ( i.e. the drawback of not being able to use them to model the progression of the neurodegenerative diseases as they are associated with aging)
This suggestion is made to address the following issues with the current version (V2) of the manuscript:
Markers of limited direct relevancy to discussed neurodegenerative diseases in the zebra fish experiments (such as neuroxin) should be reduced and the focus should be shifted to relevant markers.
Brain tissue damage and other relevant molecular tests should be included in addition to behavioral tests as behavioral tests alone are not enough to establish using zebra fish as a useful model
The heavy metal bioaccumulation section (section 3.4) should be removed with the bioaccumulation and histopathology of the mentioned experiments in previous sections added to those sections rather than made separate for better flow
Author Response
Reviewer #3 – Review Report Round 2
Your recent additions and adjustments based on the comments made by the reviewers on the manuscript are appreciated however, the title should be narrowed down to zebrafish as the majority of the manuscript discuss the uses of zebrafish as a model for neurodegenerative diseases and scarcely are other aquatic organisms mentioned.
We modified the title of the manuscript to better emphasize the use of the zebrafish. Thank you for your input.
Furthermore, the manuscript in its current form is not accepted for publication.
We endeavoured to better our latest version of our manuscript according to your input and hope we have achieved that. Thank you for your input.
I suggest a restructuring of the manuscript to the following to better fit your newly defined scope:
- An Introduction that includes the relevant common structures and conserved mechanisms in the brain between humans and aquatic animals such as zebra fish in the context of modeling human neurodegenerative diseases.
We adjusted the Introduction with a context of neurological disorders and similarities and limits of human and zebrafish brain. Thank you for your input.
- Establishing the link between heavy metal exposure and the neurodegenerative diseases.
We provided a background for heavy metals and neurodegenerative diseases and established a link between them. Thank you for your input.
- Explaining the relevant markers for each disease that exist for both humans and zebra fish (that are affected by exposure to heavy metals)
We proceeded to layout certain well-known pathological pathways for neurodegenerative diseases. Thank you for your input.
- Using the mentioned relevant markers to establish a way of using the aquatic organisms as a model for each disease (i.e. an experiment that exposes zebrafish to heavy metals and measures the accumulation of the mutated tau protein associated with dementia in the zebra fish CNS or other specific brain damage that translates to the damage caused by a neurodegenerative disease in the human CNS)
We argued the usefulness of zebrafish models with a few examples of heavy metal-induced neurotoxicity experiments to showcase the neuropathological mechanisms of neurodegenerative diseases. Thank you for your input.
- Sections that describe the experiments carried out on zebra fish including their relevant methodology and tissue sampling (for heavy metal exposure to areas relevant to the CNS function)
We revised the sections describing experimental settings according to your specifications. Thank you for your input.
- A conclusion that includes the limitations of the aquatic models ( i.e. the drawback of not being able to use them to model the progression of the neurodegenerative diseases as they are associated with aging)
We concluded our work with remarks also pertaining to the limitations of using zebrafish as a model for neurodegenerative diseases. Thank you for your input.
This suggestion is made to address the following issues with the current version (V2) of the manuscript:
Markers of limited direct relevancy to discussed neurodegenerative diseases in the zebra fish experiments (such as neuroxin) should be reduced and the focus should be shifted to relevant markers.
We directed our attention towards more relevant markers of neurodegenerative diseases in the zebrafish experiments. Thank you for your input.
Brain tissue damage and other relevant molecular tests should be included in addition to behavioral tests as behavioral tests alone are not enough to establish using zebra fish as a useful model.
A comparison section, between behavioural and other types of analysis recorded in zebrafish tests, was added to resolve this issue. Thank you for your input.
The heavy metal bioaccumulation section (section 3.4) should be removed with the bioaccumulation and histopathology of the mentioned experiments in previous sections added to those sections rather than made separate for better flow.
We moved the suggested section and believe we have achieved a better flow of the text. Thank you for your input.
Round 3
Reviewer 3 Report
Thank you for the response and for addressing the majority of the previous comments. This manuscript is adequate and i have accepted it